# African inland wetland area on the rise during the 21st century

Anzhen Li [1,2], Shengbo Chen [1] ✉, Kaishan Song[2], Yuanzhi Zhang[3,4] ✉, Zongming Wang [2] & Dehua Mao[2]

Wetlands face threats from climate change and human activities worldwide, yet the status of African wetlands remains unknown. This study mapped African wetlands and assessed area loss, drivers, and future trends under climate change using 270,000 sampling points, 810,000 Landsat images, and soil moisture data from 14 CMIP6 models. The results reveal no large-scale loss of wetlands in Africa from 1984 to 2021 (0.51% net loss), with the loss concentrated in coastal areas (9.64% net loss), while inland wetlands show a slight increase in area (0.50% net increase). A comparison of the time series of wetland area and related drivers showed that the change of inland wetland area is closely related to climate change, and human activities have exacerbated the loss of coastal wetlands. TOPMODEL projections suggest an upward trend in inland wetland area by 2100, but uncertainty persists and inland wetlands remain at risk of loss in the future.

Wetlands are critical to achieving global commitments related to biodiversity, climate change, and sustainable development[1]. Since the signing of the Ramsar Convention in 1971, protecting wetland ecosystems has been officially recognized as an international priority[2,3]. The Aichi target set by the Convention on Biological Diversity in 2010[4] and Sustainable Development Goal target 6.6.1 of the 2030 Sustainable Development Agenda[5], along with the Paris Climate Agreement, are closely linked to wetlands[6]. Despite the widespread distribution of wetlands around the world, some evidence indicates that wetland areas are rapidly decreasing[7–9]. According to the latest research, global wetlands have decreased by 21% since 1700. This wetland loss has been concentrated in Europe, the United States, and China and has expanded rapidly during the mid-twentieth century[2]. Africa remains the least developed region in the world, and its wetland status and loss remain a matter of debate[10]. Whether large-scale losses of African wetlands have occurred in the past, what drives them and how they will change in the future have always been topics of interest. Understanding the value of wetlands and the ongoing changes they are experiencing is essential to ensure their protection and rational use[11,12]. Accordingly, we conducted the remote sensing mapping of African wetlands using an approach

that combined high spatial resolution and long-term series over the past 38 years (1984–2021). From the data collected, we systematically analyzed and studied three aspects of African wetlands: historical loss, change drivers, and wetland change under future climate change.

Following the definition of wetlands presented in the Ramsar Convention on Wetlands[3], this study conducted a consistent analysis of the distribution of and changes to African wetland ecosystems in eight subcategories of two major categories—inland wetlands and coastal wetlands (Supplementary Fig. 1 and Table 1)[10]. We began by using remote sensing methods to classify and map African wetlands from 1984 to 2021[13,14]. We extracted African wetlands as systematically and accurately as possible by evenly arranging as many as 270,000 sampling points (Supplementary Fig. 2) at intervals of 0.1° (latitude) × 0.1° (longitude) throughout Africa. We then confirmed the wetland type of each individual sampling point through visual interpretation. Based on these sampling points, we used the random forest classifier to classify and extract African wetlands in 9 periods from 1984 to 2021 on the Google Earth Engine (GEE) platform. After completing this high-resolution mapping of African wetlands in the historical period (1984–2021), we counted the time series of African

[1]College of Geo-Exploration Science and Technology, Jilin University, Changchun, China. [2]State Key Laboratory of Black Soils Conservation and Utilization, Northeast Institute of Geography and Agroecology, Chinese Academy of Sciences, Changchun, China. [3]School of Marine Sciences, Nanjing University of Information Science and Technology, Nanjing, China. [4]Department of Architecture and Civil Engineering, Faculty of Engineering, City University of Hong Kong, Hong Kong, China. ✉e-mail: chensb@jlu.edu.cn; yzhang209@nuist.edu.cn

wetland area and used them for a comparative analysis with the time series of climate change (including temperature, precipitation, drought, and soil moisture)[15–18] and human activities[19,20] to assess the impact of different drivers on wetland changes[21]. To further investigate the loss of wetlands in Africa under future global climate change, we used CMIP6 climate data[22,23] and the TOPMODEL-based diagnostic model[24] to predict and analyze changes in inland African wetlands by 2100 based on the classification results of historical periods (Supplementary Fig. 3). The TOPMODEL method, the most popular model for wetland simulation, has been widely applied to downscale mean water table depth in a catchment or grid scale to wetland area based on local topography distribution[25–27]. The present study used the SM data produced by 14 models of the CMIP6 project (Supplementary Table 2) as the input for its discussion of the changes in the area of inland wetlands in Africa under the four shared socioeconomic pathway (SSP126, SSP245, SSP370, and SSP585) scenarios by the end of the current century (more details in the "Methods" section).

## Results and discussion

### Wetland variation in Africa from 1984 to 2021
Our results reveal that no large-scale loss of African wetlands occurred on the whole from 1984 to 2021. The possibility of a huge loss of more than 50% of global wetlands[28] has been met with skepticism in light of the fact that the phenomenon occurs only in certain wetland categories in some specific regions[29], such as North America, Europe, and Asia, and cannot be extrapolated to the broader geographic regions or other wetland categories[9,30]. Africa represents an important part of the world's wetland ecosystem, leading to intense scholarly interest in the possibility of large-scale losses over the past 38 years[31]. However, our study found that from 1984 up to date, although Africa had lost about 138,500 km² of wetlands, these losses were offset by a gain of 132,400 km² (Supplementary Fig. 4), indicating that Africa's wetlands, even though slightly reduced in overall area, have not experienced loss on a large scale. According to our statistics, the total area of African wetlands in 1984 was 1,200,800 km² (excluding shallow marine water) and net loss area was about 6100 km² over 38 years, accounting for only 0.51% of the wetland area in 1984 (Supplementary Table 3).

The loss of wetlands in Africa varies with their wetland category and zonality (Fig. 1). Among all eight wetland categories, only inland surface water, inland salt pan, and shallow marine water areas increased, demonstrating gains of 3.82%, 5.80%, and 12.28%, respectively (Here and the subsequent wetland area change were net change). In contrast, all vegetation-covered wetland categories experienced area loss over the same time span. The two categories with the largest area loss over the 38-year period were inland swamps and coastal marshes, which showed a net loss of 6684.40 and 4406.23 km², respectively. Among the wetland categories where loss occurred, we noted a much higher rate of loss in coastal wetlands compared to inland wetlands in Africa: specifically, tidal flats were decreased by 12.2%, coastal swamps reduced by 9.50%, and coastal marshes declined by 8.65%. As for inland wetlands, the areas of inland swamps and inland marshes decreased by 1.60% and 0.22%, respectively. We also found an interesting phenomenon when reviewing the statistics at the country level in that the countries featuring increased wetland area were mainly concentrated in north of the equator, while the wetland area of southern African countries decreased to some extent (Supplementary Fig. 5). Among all countries, the Democratic Republic of the Congo exhibited the most serious wetland loss, a decrease of about 12,000 km² from 1984 to 2021, which could be mainly attributed to the loss of forest wetlands in the Congo Basin[21,32]. Additionally, the entire southern part of Africa (south of 10°S latitude) demonstrated a considerable wetland loss and little wetland growth compared to the other parts of Africa.

The 38-year time span allowed us to study African wetlands in terms of time series and trends. Accordingly, we discovered different trends when comparing changes in the areas of coastal wetlands and inland wetlands in Africa from 1984 to 2021 (Fig. 2 and Supplementary Fig. 6). The coastal wetland (excluding shallow marine water) lost 11,500 km² in area over 38 years, comprising nearly 9.64% of the whole, and its area change time series revealed a completely linear decline trajectory[29]. This observation indicates that the coastal wetland ecosystem, which is at the same time one of the most productive and highly threatened systems in the world, has been severely damaged in Africa, as in the other areas of the globe, which is basically consistent with the conclusions of other studies[13,14,33]. Compared with the large-scale loss of coastal wetlands, the time series of inland wetlands displayed a slight upward trend, with a net increase of 0.50% in area. On the whole, from 1984 to 2021, African wetlands did not increase or decrease linearly but followed a strange trajectory that can be characterized as slightly decreasing overall, followed by a significant decline in a specific year (the middle year 2005, 2017) and then recovery to some extent. We believe that the large-scale loss of coastal wetlands within the 38-year period has offset the increase in the area of inland wetlands, resulting in a slight decline in the total area of wetlands in Africa as a whole[34]. It should be noted that due to missing remote sensing imagery and cloud contamination, the classification results of this study differ from existing datasets such as GIEMS, WAD2M, and GWL-FCS30. Consequently, current research conclusions –particularly for periods before 2000–remain subject to uncertainty. A detailed comparison between our results and these datasets, along with explanations of the differences, is provided in the Supplementary Information.

### Driving forces for African wetland variation
Global coastal wetlands are highly threatened by human activities, and Africa is no exception[14,29]. This study used the Human Footprint dataset (HFP) to assess the intensity of human disturbance in the coastal and inland wetland regions of Africa from 2001 to 2020[20,35]. Based on our analytical results, within those two decades, the average HFP in the coastal wetland area of Africa was 13.46, while that in the inland area was 6.79, indicating that the degree of human disturbance in the coastal wetland area was approximately twice as much as that in the inland wetland on average (Supplementary Fig. 7). It is worth noting that HFP in both coastal and inland regions grew at a high rate. Compared with 2000, HFP in inland and coastal regions of Africa increased by 11.02% and 11.91%, respectively, by 2020 (Supplementary Fig. 8). Population growth and increasing economic development have caused a certain degree of damage to coastal wetlands around the world (including Africa). The primary direct driver for the loss of coastal wetlands has been converted to other land uses (aquaculture, agriculture, coastal developments, etc.), which is the general consensus of the international community[29,33,36]. Although human interference contributed less to changes in inland wetlands in Africa, the signs of human activities are increasing year by year, demanding increased public attention. Of course, in addition to the interference posed by human activities, sea-level rise, storm erosion, and other unique drivers in coastal areas have caused some damage to wetlands, comprising one of the reasons why coastal wetland losses are higher than those of their inland counterparts[37–40].

Climate change also has had a certain impact on African inland wetlands. In this study, we compared the correlation between climate drivers (temperature, precipitation, drought, and soil moisture) and wetland area changes in Africa over the past 38 years from the perspective of time series (Fig. 3)[41]. As a key indicator of climate change in Africa[42], the temperature increased almost linearly by about 1° from 1984 to 2021, but this change in temperature did not bring about a corresponding linear increase or decrease in wetland areas in Africa. Interannual precipitation in Africa exhibits substantial uncertainty across different data products[43], and the spatiotemporal patterns of precipitation from 1984 to 2021 are not consistent at the continental

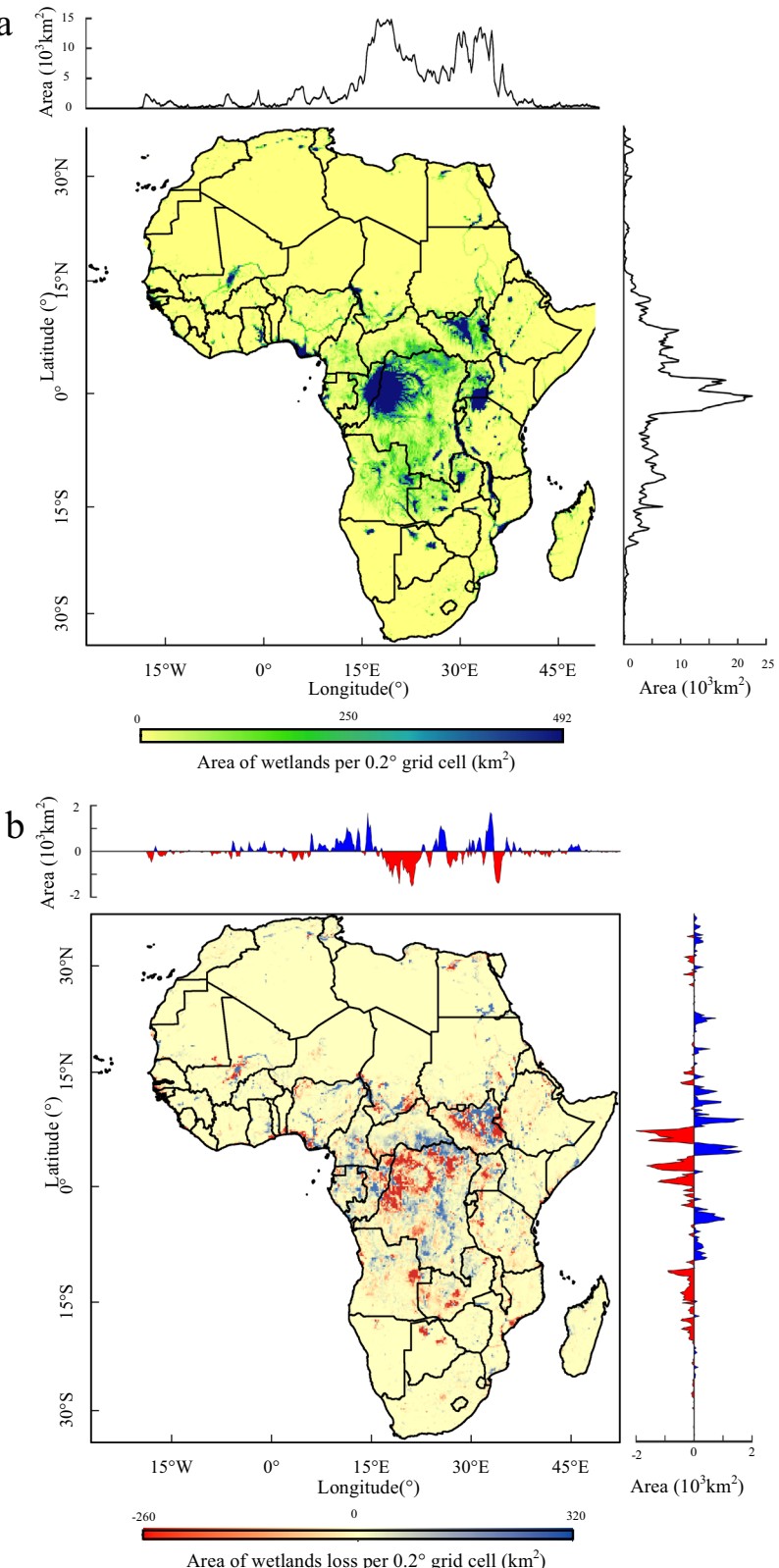

**Fig. 1 | Area distribution and changes of African wetlands during the historical period. a** The distribution of wetland areas in Africa from 1984 to 2021 (The average wetland area in 9 periods). **b** The distribution of wetland area changes between the two periods of 2019–2021 and 1984–1990. The wetland loss is shown in red and the gain is shown in blue. The map is drawn according to a 0.2° × 0.2° grid. The upper and right sides show the area changes of wetlands in longitude and latitude.

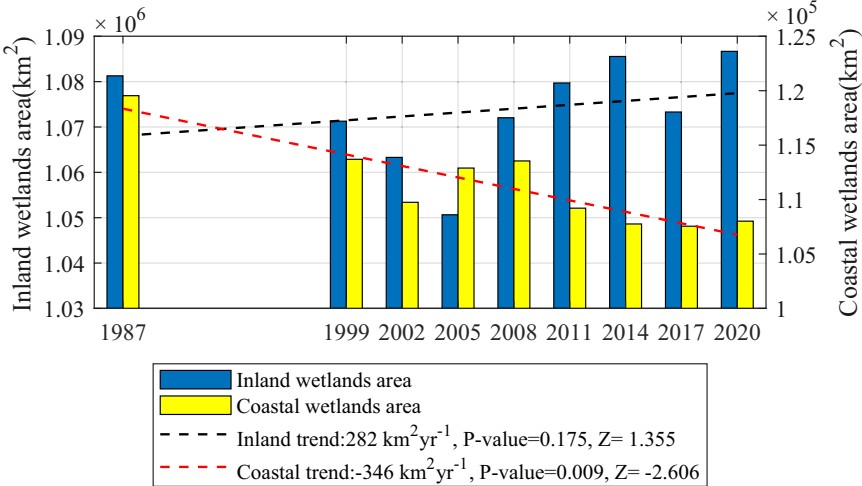

**Fig. 2 | Changes and trends in the area of inland and coastal wetlands in Africa during the historical period.** The black dotted line is the inland wetland change trend, and the red dotted line is the coastal wetland change trend. Trend lines are obtained by one-dimensional linear fitting, and trend significance is assessed using the Mann–Kendall nonparametric trend test; the test statistic Z and two-sided P-values are reported (two-sided test). The horizontal coordinates were the middle years of 9 classification periods.

scale. Notably, decadal-scale variability plays an important role in explaining precipitation trends and the episodic (phase-like) changes in wetland extent: the episodic increases and decreases of wetland area in some regions can be partly attributed to internal climate oscillations on decadal or longer timescales rather than to sustained linear trends. Although some regions show moderate correlations between wetland area and precipitation, this relationship is highly spatially heterogeneous and temporally non-stationary (see Supplementary Information). Moreover, hydrological processes such as evapotranspiration, surface runoff, and groundwater recharge also modulate the effect of precipitation on local wetlands[44–46]; therefore, integrated, region-specific analyses are required. As one of the extreme climate phenomena caused by climate change, drought had little impact on African wetlands as a whole yet showed a high degree of consistency with wetland changes locally[18,47]. The forest wetlands in the Congo Basin have been on a downward trend in recent years, which is connected to the continuous drought in the region (Supplementary Fig. 9)[21]. Soil moisture, as a response factor to the drivers of climate change such as temperature, precipitation and evapotranspiration, reflects the state of wetlands to some extent, and is the most relevant climate factor to wetland changes[24]. The loss and subsequent recovery of African wetlands before 2005 and the increase in area after 2017 were consistent with the changes in soil moisture.

The above analysis uncovers a strong correlation between inland wetlands and climate drivers. Of course, analyzing the impact of climate change on wetlands only from the perspective of time series was rather one-sided, not only because of the complex relationship and feedback between climate (precipitation, temperature, extreme climate, etc.) and wetlands but also because of the possible synergistic effect of climate drivers and human factors[37]. Our work demonstrates that continuous monitoring, experiments, and simulations are still needed to understand the drivers of wetland changes to fundamentally solve the problem of wetland loss.

## Inland wetland trends under future climate trajectories

In the 21st century, the annual average temperature in some regions of Africa may rise at about 1.5 times the rate of global temperature increase[23,48], while the frequency and intensity of precipitation remain more uncertain[49–51]. Global climate change is expected to exacerbate the loss and degradation of many wetlands. However, predictions of this loss have not been quantified[29]. In this study, the CMIP6 SM data and the TOPMODEL-based diagnostic model were used to simulate and analyze changes in African inland wetlands under the influence of climate change in the future (to 2100)[23,24] (more details in the "Methods" section). According to our simulation results, the area of African inland wetlands shows an upward trend over the next 70 years under all of the SSP scenarios, whether SSP126, SSP245, SSP370, or SSP585 (Fig. 4). Even though the soil moisture data of the 14 models vary greatly, and with the extension of the simulation time, the uncertainty of the simulation results was also increasing. However, most of the models showed that the impact of climate change on inland wetlands tended to be positive, and the average wetland area based on the 14 models increased steadily under the four SSP scenarios (Supplementary Fig. 10).

Similar to the changes observed in historical periods, African inland wetlands will also exhibit different degrees of loss in area over the next 70 years, but the losses will be offset by higher gains (Supplementary Fig. 11). Based on the average of the results of 14 simulations, it is found that the inland wetland area will have a net increase of more than 10% under the SSP126/245/370/585 scenarios from 2023 to 2100. The future growth areas of inland wetlands are mainly concentrated in northern Africa, especially in the sub-Saharan Sahel region. In the SSP126 scenario, which features more moderate climate change than the others, 17% of Africa's regions will experience wetland area growth; in comparison, the SSP585 scenario predicts that 35% of the regions will experience wetland area expansion (Fig. 5).

The results of the 14 models show a positive trend, but uncertainty still persists and a risk of future wetland loss remains (Fig. 6). In particular, even under the SSP126 scenario of green sustainable roads, the simulation results point to a net loss of about 63,000 km² of wetlands in Africa. These losses can be severe, permanent and with negative implications of specific wetland ecosystem types that cannot be "offset" by gains elsewhere. From the perspective of spatial distribution, the future areas of wetland loss in Africa are located mainly in the Congo Basin and western Africa; comparatively, the loss of wetlands in North Africa (Mediterranean coast) is also serious, although the region does not have much wetland distribution. It is worth noting that the loss of forest wetlands in the Congo Basin is at risk of further aggravation. The simulation of the four SSP scenarios illustrates a downward trend in Congo Basin, which is basically consistent with the trend of the

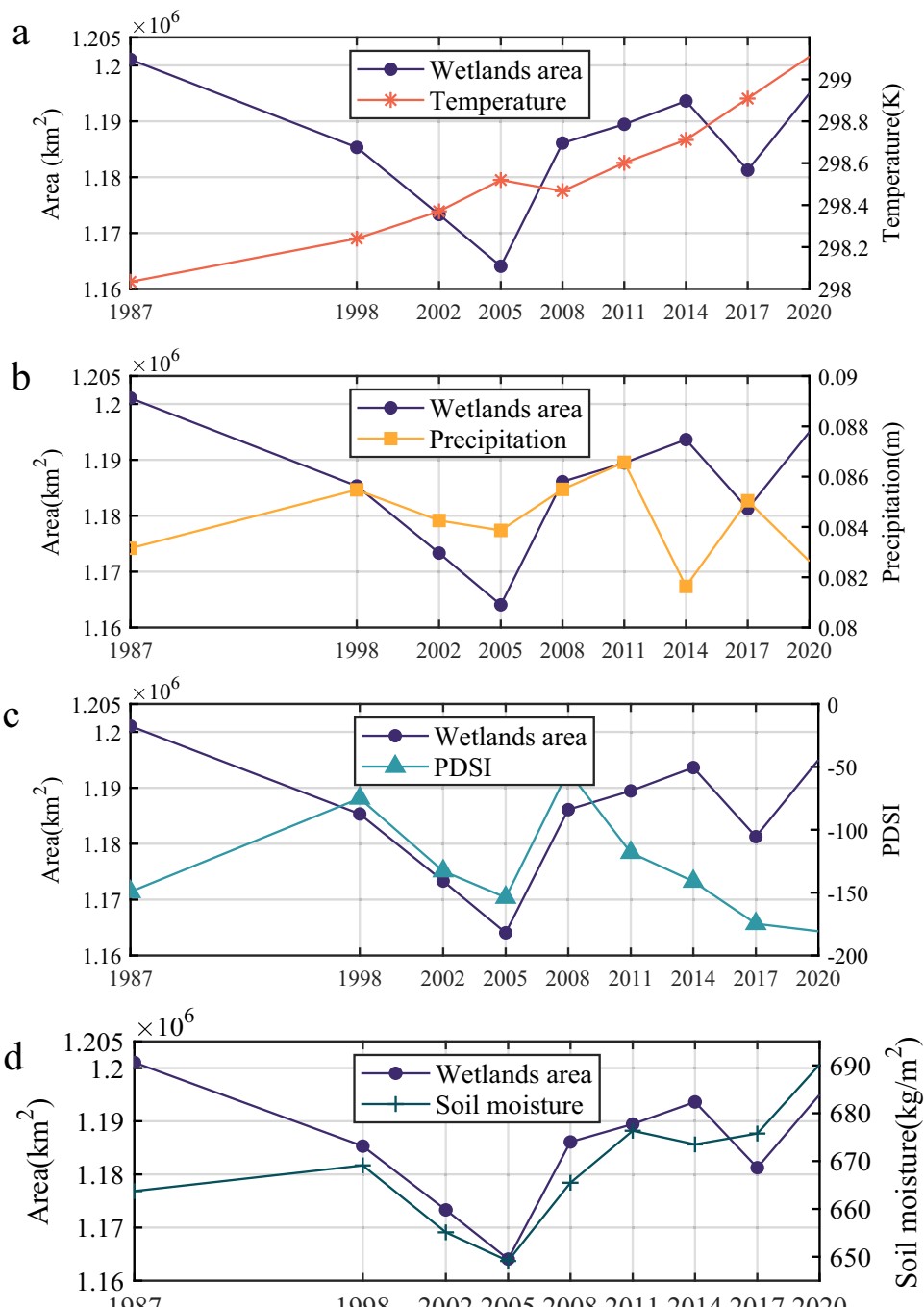

**Fig. 3 | Time series comparison of wetland area and climate drivers in Africa from 1984 to 2021. a–d** The monthly average air temperature at 2 m height (K), monthly total precipitation (m), palmer drought severity index (PDSI), and 0–200 cm underground soil moisture (kg/m²) in the main distribution areas of African wetlands.

classification results in the historical period and is also consistent with previous research findings[21,32]. In the SSP370 and SSP585 scenarios featuring more severe levels of global climate change, the Okavango Delta, Etosha National Park, and Zambia National Park in southern Africa are also at risk of wetland loss.

Human activities, climate change, and other factors jointly drive future changes in African wetlands; conversely, wetland changes will have both direct and indirect impacts on human well-being. In recent years, the COVID-19 pandemic has fundamentally people's perceptions of health and the environment. It has been realized that unsustainable use and inappropriate management of wetlands can extend beyond the loss of ecosystem services to include direct risks, including diseases[1]. As a vital component of the world's wetland ecosystems, the

relatively well-preserved wetland ecosystems in Africa (compared to countries or regions such as Europe, China, and the United States[2]) have made critical contributions to global biodiversity and climate change in the past and will play an increasingly important role in the future. In addition, as the most easily reclaimed and cultivated land type, wetlands have played a vital role in the livelihood of African people for thousands of years[52]. It is important to note that Africa's wetlands have not experienced large-scale overall loss, but there has been substantial reduction in certain regions (such as the Congo Basin, Southern Africa) and in coastal wetlands. The quality of Africa's wetlands still needs to be assessed, as stable or expanding wetland areas do not necessarily indicate a favorable condition. The degradation of the ecological functions of existing wetlands continues to pose serious

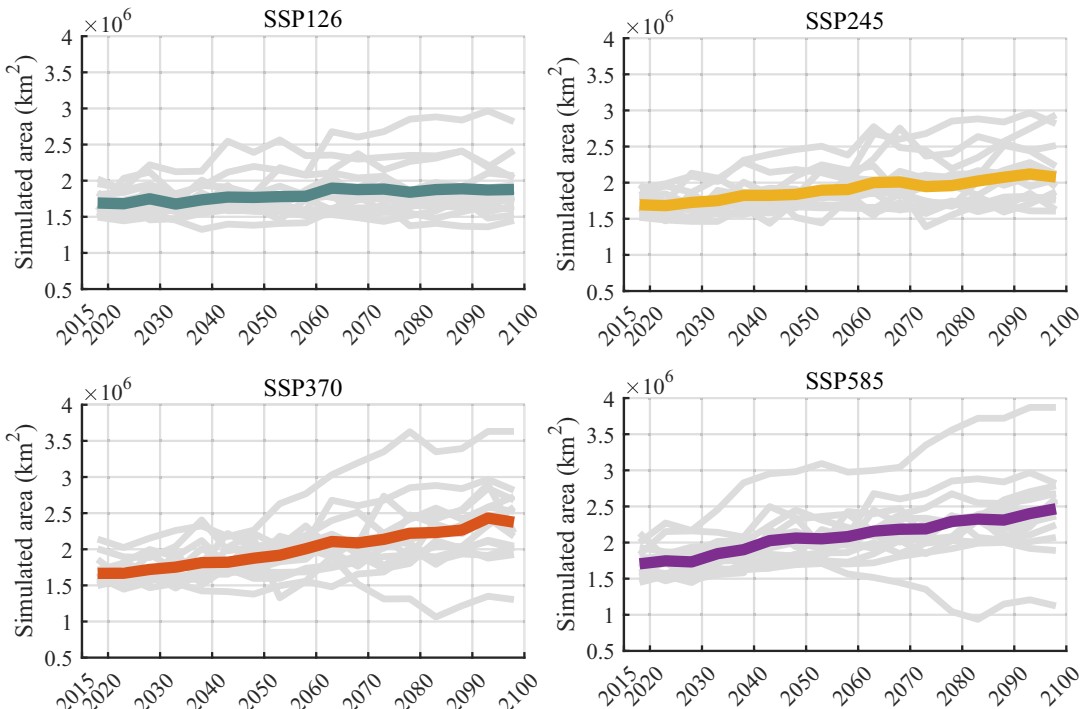

**Fig. 4 | Trends in the area of African inland wetlands by 2100 under four SSP scenarios.** The gray lines are the time series of inland wetland area of 14 models, and the color lines are the average of 14 models. The horizontal coordinates are the simulation time (year) and the vertical coordinates are the simulated wetland area in square kilometers.

threats to local communities, such as the shrinking of Lake Chad's surface area. The future expansion of Africa's wetland area offers the potential to provide food and water for more people and alleviate poverty in remote rural areas. However, it must be realized that the simulation of this study is only an assessment of the future trend of African inland wetlands. The impact of extreme weather (floods, droughts, La Niña phenomenon, etc.) and the direct destruction of wetlands by human activities make future wetland changes highly uncertain. Therefore, we suggest that African government departments should remain sensitive and vigilant and consider these findings in future planning, including disaster management planning and risk assessment, so that African wetland ecosystems can play a key role in responding to the decisive challenges of our era.

Using high spatial resolution maps of African wetlands over the past 38 years, this study analyzed the drivers of wetland loss and change in Africa. Based on the analytical results, future changes in African inland wetlands were further predicted under different SSP scenarios. Our findings support the following conclusions: (1) There was no large-scale loss of African wetlands from 1984 to 2021. The losses were concentrated in the coastal areas, the Congo Basin and southern Africa. (2) The primary drivers of changes in coastal and inland wetlands in Africa were different. Climate change was closely related to the change of inland wetlands, while human activities had a greater impact on coastal wetlands. (3) There was a trend toward the further expansion of inland wetlands over the next 70 years, but uncertainties still exist and a risk of future wetland loss remains.

An in-depth understanding of wetlands is fundamental to their conservation and utilization. Even in the context of rising wetland area, irrational human activities (e.g., damming, reclamation, and irrigation) can still bring humanitarian disasters to residents, and the focused conservation of Africa's wetlands remains urgent and necessary. Importantly, governance should pursue integrated strategies that protect wetland functions while enabling sustainable economic growth and improved livelihoods; our findings are not intended to discourage development, but to inform development pathways that are compatible with wetland conservation and community resilience. We hope that our work will contribute to improving the management and conservation of African ecosystems and, in the process, help support those who depend on these ecosystems.

## Methods

### Wetland classification system

Africa, the world's second largest continent, accounts for about one-fifth of the total land area of the earth[53]. It was estimated that the global wetland area was nearly 9.2 million square kilometers, of which 1.3 million square kilometers were located in Africa[12,54,55]. Due to the different definitions and mapping methods of wetlands, existing studies have different estimates of the spatial extent of wetlands around the world, including Africa[56].

This study's investigation was based on the definition of wetlands proposed by the Ramsar Convention on Wetlands, as "areas of marsh, fen, peatland, or water, whether natural or artificial, permanent or temporary, with water that is static or flowing, fresh, brackish or salt, including areas of marine water the depth of which at low tide does not exceed 6 m." Accordingly, we proposed a wetland remote sensing classification system in line with African realities. African wetlands are divided into two categories, inland wetlands and coastal wetlands. Specifically, inland wetlands include the sub-categories inland surface water, inland marsh, inland swamp, and inland salt pan. Coastal wetlands include the sub-categories coastal marsh, coastal swamp, tidal flat, and shallow marine water (Supplementary Table 1)[10]. This study only distinguishes natural wetlands since paddy fields, for example, are less widely distributed in Africa and are more suitable for classification as farmland[57].

The scope of this study is the entire African continent and its affiliated islands, including 61 countries or regions, with an area of

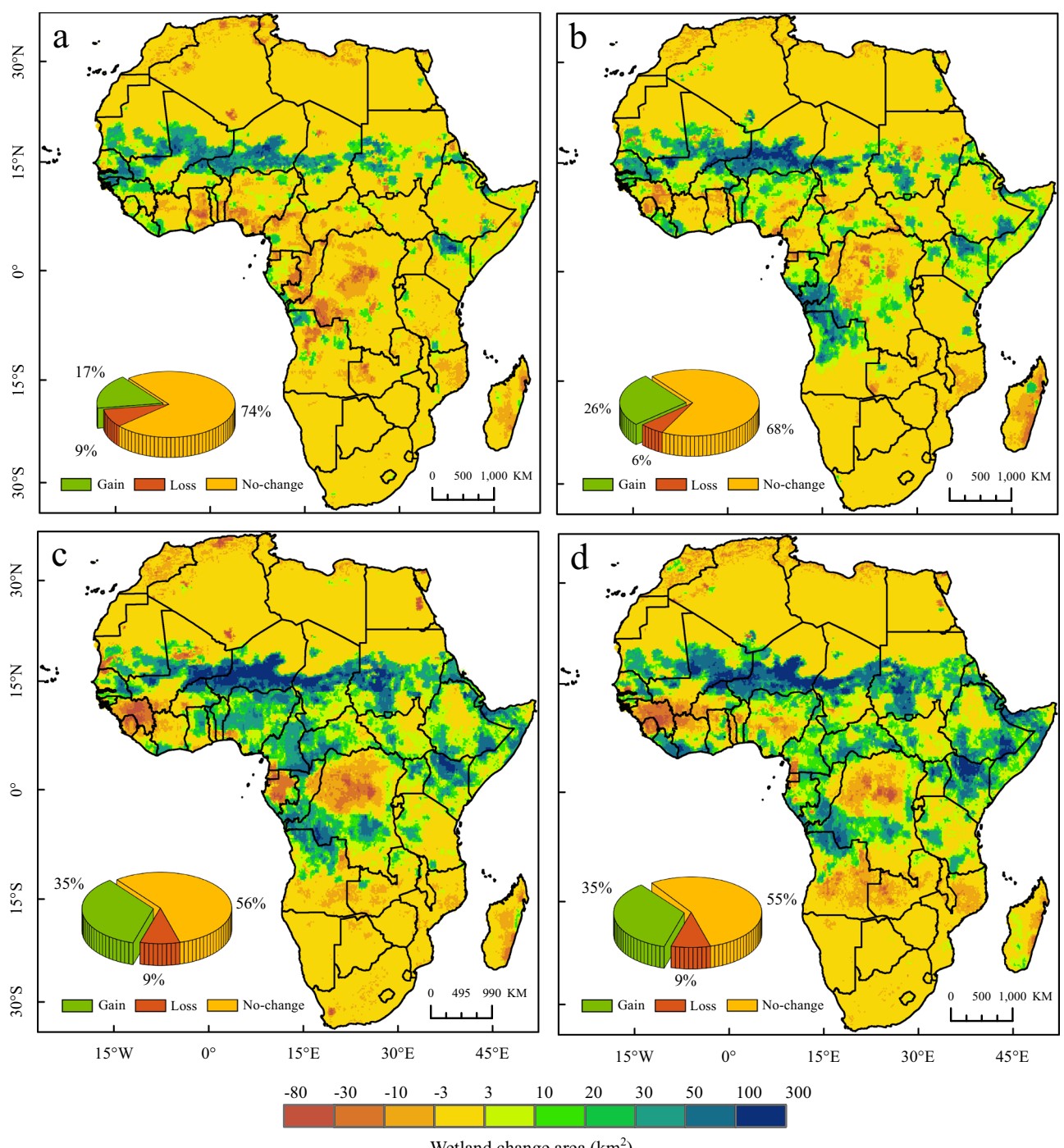

**Fig. 5 | Inland wetland area changes in Africa by 2100 under four SSP scenarios mapped per 0.2° grid cell. a–d** Inland wetland changes under SSP126, SSP245, SSP370, and SSP585 scenarios, respectively. The pie chart shows the proportion of areas with inland wetland area changes in Africa.

about 30 million km². The specific boundaries used in the study were determined based on the boundaries of African countries in the large-scale international boundary (LSIB) dataset. Based on this boundary, the coastal area boundary was extended to a depth of 6 m. The water depth data were drawn from the global relief model of Earth's surface (ETOPO1)[58] (Supplementary Fig. 1).

### Remote sensing classification method

In this study, the wetland remote sensing classification and mapping work in the historical period of Africa were mainly divided into four parts: image preprocessing, wetland sampling, feature construction,

and classification mapping (more details in the Supplementary Information).

Remote-sensing image preprocessing included standard cloud-and-shadow masking and noise removal. We applied robust annual compositing (e.g., multi-temporal median composites) to suppress short-term cloud contamination. To evaluate the effectiveness of these procedures, we compared our outputs against independent, cloud-insensitive reference products (for example, GIEMS). These comparisons indicate that, owing to the substantial improvement in Landsat data coverage and revisit frequency after 2000 and to our compositing plus multi-dataset validation strategy, the wetland time series constructed for the period after 2000 show no evidence of a systematic

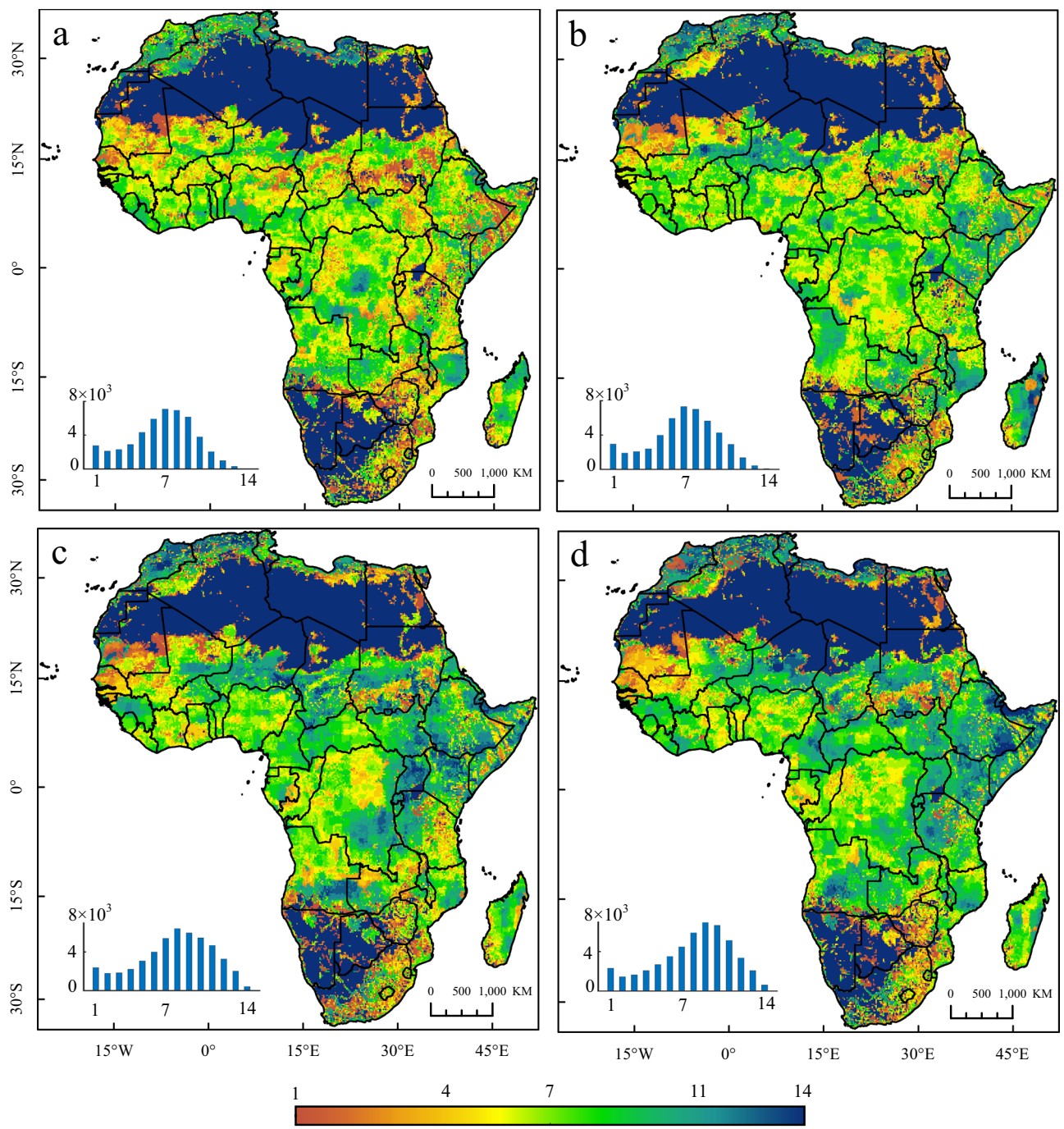

**Fig. 6 | The uncertainty of the spatial distribution of the area change of African inland wetlands by 2100. a–d** Number of the models agreeing on wetland loss, increase and unchanged in the 14 models under the SSP126, SSP245, SSP370 and SSP585 scenarios. The bar chart is the statistics of the number of grids corresponding to the uncertainty (excluding unchanged regions such as the Sahara Desert); the horizontal coordinates is the number of the models agreeing on wetland change, ranging from 1 to 14; the vertical coordinates is the number of 0.2 ° grid cells.

bias attributable to cloud contamination. By contrast, data before 2000 are affected by larger image gaps and persistent cloud cover, resulting in reduced sample size and representativeness and therefore greater uncertainty in quantitative trend estimates. Relevant comparison figures and statistical summaries are provided in the Supplementary Information.

The random forest method in GEE was used to classify wetlands[59]. This algorithm has various advantages, including high accuracy, the ability to effectively run on large datasets, and the capability to evaluate the importance of each feature in classification problems; in addition, the algorithm does not over-fit the training samples[60]. Before performing formal classification, several experimental areas were selected in Africa, such as the Congo Basin, the Sudd Wetland in South Sudan, and the Okavango Delta, using different supervised classification classifiers and a variety of wetland classification feature combinations for classification experiments. Finally, random forest

classification was identified as meeting our research requirements with its fast and accurate characteristics. In the process of classification, the sampling points obtained by visual interpretation were randomly assigned in a ratio of 7:3, meaning that 70% of the sampling points were used to train the classifier, while the remaining 30% were used for accuracy verification. Since the study area covered an area of about 30 million km² across Africa, coupled with a large number of sampling points and training datasets, it was impossible to classify and extract wetlands throughout Africa at one time, even in GEE. Therefore, the entire African land mass was divided into 22 sub-regions, and each region was classified separately, after which the results were spliced to obtain a complete African wetland classification result.

### Area statistics and time series

The wetland area involved in this study was based on the wetland classification map. A total of 9 wetland maps were generated from the classification of African wetlands in historical periods (more details in the Supplementary Information). The area of all wetland pixels was accumulated in each of the wetland maps with a spatial resolution of 30 m in the GEE platform to obtain the wetland area of each of the 9 periods.

The study's time series of drivers, such as climate and human activities, also entailed calculating according to the same time and spatial resolution as the classification. The data regarding temperature, precipitation, PDSI, SM, and HFP were uniformly resampled to a resolution of 30 m, whereupon the mean values of the main distribution areas of wetlands in 9 periods were calculated. Ultimately, the time series consistent with the spatial and temporal resolution of wetland area was obtained, which we then compared with the wetland time series to analyze the correlation and consistency. The trend lines in the figures of the paper were obtained via one-dimensional linear fitting based on the time series data, and the $R^2$ value was given.

### Human activities and climate data

The study used the Human Footprint to assess the overall impact of human activities on wetlands. The HFP is a composite index calculated as a weighted sum of eight classes of human-pressure maps, including maps of population density, infrastructure, human accessibility, and energy supply. The higher the value, the greater the human pressure on the environment[19].HFP data comprise the most widely used index to measure human impact in the world and show the gradient of human impact. The HFP map space used in this study has a total of 21 maps, with a resolution of 1 km, from 2000 to 2020[61].

In order to evaluate the climate drivers affecting wetland changes, we used climate data, such as temperature, precipitation, drought index (PDSI), and SM, for our analysis. The temperature and precipitation data were from the ERA5 monthly comprehensive data provided free of charge in GEE, meaning the average temperature at 2 m height (monthly average) and total precipitation (monthly sum). ERA5 refers to the fifth-generation atmospheric reanalysis data of the European Centre for Medium-Range Weather Forecasts (ECMWF) for global climate. The drought index was derived from the Palmer Drought Severity Index (PDSI) data in TerraClimate data[15]. TerraClimate is a global dataset featuring monthly land surface climate and climate water balance data with a resolution of 4638.3 m. PDSI values range from −4317 to 3418; the smaller the value, the higher the degree of drought. SM data include both actual observed data and simulated data from CMIP6 (more details in the Supplementary Information).

### Future wetland simulation

The simulation process of African inland wetlands in the future period was mainly divided into three stages (Supplementary Fig. 3). The first stage was the remote sensing classification and mapping for wetland of the historical period (1984–2021). The 10-period wetland maps produced in this stage, and the subsequent analysis of change drivers,

showed that the inland wetlands in Africa were closely related to the climate, and climate data could be used to predict future changes in inland wetlands. The classification results of the historical period were also used to determine some parameters (such as M) in the TOPMODEL-based diagnostic model and to verify the subsequent simulation results.

The second stage was the inland wetland simulation of historical periods (1984–2021), that is, TOPMODEL-based diagnostic model with soil moisture from the FLDAS dataset. The main purpose of this simulation was to determine the specific values of some parameters in the simulation process and verify the accuracy of the simulation method. Different from soil moisture data simulated in CMIP6, soil moisture data obtained from FLDAS was used as the real value of soil moisture in this study. The WTD was first calculated based on FLDAS soil moisture data (Equations (4)), and then calculated the average WTD and CTI of more than 40,000 basins in Africa. Since the range of M is set to 1–15, the CTI threshold from 15 values of M is generated (Equations (2)), and then 15 corresponding African simulated wetland areas are obtained. According to the classification results of historical periods, the RMSE corresponding to 15 M values was calculated, and then determined the M value of 40,000 watersheds based on the minimum RMSE. After calculating the CTI threshold using the best fitted M value, the TOPMODEL simulated wetland area based on FLDAS soil moisture for 9 periods was finally obtained.

The third stage was the inland wetland simulation of the future periods (2015–2100), that is, TOPMODEL-based diagnostic model with soil moisture from the CMIP6 dataset. Similar to the previous simulation process, the WTD was calculated under four future scenarios based on the soil moisture data of 14 CMIP6 models, and then the inland wetland area from 2015 to 2100 in four SSP scenarios was obtained. In addition, in order to better demonstrate the long-term sequence changes of African wetlands, the inland wetland area changes were also simulated during the period 1984–2014 based on CMIP6 soil moisture data under the "historical" scenario, and compared the differences between the historical period simulation based on CMIP6 data and the actual observation results (remote sensing classification) (Supplementary Fig. 12). More details of the product evaluation are available in the supplementary information.

### Wetland classification accuracy assessment

In order to make the extraction of wetlands as accurate and detailed as possible, about 270,000 visual interpretation sampling points were arranged throughout Africa, of which 30% of the sampling points were not used for the training of random forest classifiers, but for the accuracy verification after classification as independent verification points, that is, the calculation of confusion matrix. The results of the remote sensing error matrix showed that the wetland maps of 9 periods from 1984 to 2021 had high accuracy. Moreover, with the passage of time, the quality and quantity of Landsat images have improved, the classification accuracy has also increased slightly (Supplementary Fig. 13 and Table 4). On the whole, the average producer accuracy of wetlands in 9 periods was 80.76%, and the average user accuracy was 91.77%. Even in the poor image quality of 1984–1990, the accuracy of wetland producers and users can reach 77.97% and 91.52%. For the subdivided wetland categories, surface water, coastal swamp, and tidal flat were wetland categories with high classification accuracy due to their obvious and unique remote sensing characteristics, with an average accuracy of more than 85%. The classification accuracy of herbaceous marshes and inland salt marshes was relatively low, about 70%. More details of the product evaluation are available in the Supplementary Information.

### Reporting summary

Further information on research design is available in the Nature Portfolio Reporting Summary linked to this article.

## Data availability

All data used in this study are freely available from public repositories. The Landsat images used in this study are available from the US Geological Survey (http://earthexplorer.usgs.gov) and Google Earth Engine (https:// earthengine.google.com). The data of African national boundaries and 6-meter water depth boundaries are available from https://developers.google.com/earth-engine/datasets/catalog/USDOS_LSIB_SIMPLE_2017 and https://developers.google.com/earth-engine/datasets/catalog/NOAA_NGDC_ETOPO1. Human Footprint dataset (HFP) can be accessed freely at the figshare repository (https://doi.org/10.6084/m9.figshare.16571064). Temperature data are available from https://developers.google.com/earth-engine/datasets/catalog/ECMWF_ERA5_MONTHLY. Precipitation data are available from https://developers.google.com/earth-engine/datasets/catalog/IDAHO_EPSCOR_TERRACLIMATE, https://developers.google.com/earth-engine/datasets/catalog/NASA_FLDAS_NOAH01_C_GL_M_V001, and https://www.ncei.noaa.gov/data/global-precipitation-climatology-project-gpcp-monthly/access/. PDSI is available from https://developers.google.com/earth-engine/datasets/catalog/IDAHO_EPSCOR_TERRACLIMATE. Soil moisture is available from https://developers.google.com/earth-engine/datasets/catalog/NASA_FLDAS_NOAH01_C_GL_M_V001. CTI data is available from https://catalogue.ceh.ac.uk/documents/6b0c4358-2bf3-4924-aa8f-793d468b92be. Africa watershed vector data is available from https://developers.google.com/earth-engine/datasets/catalog/WWF_HydroSHEDS_v1_Basins_hybas_8. CMIP6 data is available from https://esgf-node.llnl.gov/search/cmip6/. The wetland maps for ten historical periods and the wetland simulation results for future periods produced in this study have been deposited in the Zenodo database and are provided as open data (https://doi.org/10.5281/zenodo.17865977).

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

## Acknowledgements

This research was jointly funded by the National Key Research and Development Program of China (2020YFA0714103), which supported S.C. and A.L.; the National Natural Science Foundation of China (42494821), which supported A.L. and D.M.; and the Science and Technology Development Program of Jilin Province (YDZJ202501-ZYTS579), which supported A.L.

## Author contributions

A.L., S.C. and Y.Z. conceptualized the project, acquired funding. A.L. developed wetland classification, conducted the model simulations and analysis with support from D.M., K.S. and Z.W., and drafted the manu-script with contributions from all co-authors. S.C., K.S., Y.Z., Z.W. and D.M. participated in the discussion and analysis of the results and edited the manuscript. All authors reviewed the results, revised, and approved the manuscript.

## Competing interests

The authors declare no competing interests.
