## [Transparent Peer Review file · Nature Communications]

African inland wetland area on the rise during the 21st century

Corresponding Author: Professor Shengbo Chen

Version 0:

Reviewer comments:

Reviewer #2

(Remarks to the Author)

- The overall observation of a rise in wetland coverage in the future and the downplaying of wetland loss in the past goes against a lot of independent reports and academic articles. It is well-documented that population explosions and the increasingly drying of the African continent will have dire consequences on the environment and wetlands. The authors need to ensure rigorous error analysis (i.e. error propagation and how it influences the rate of change over the time series before future projection; and the error linked to the TOPMODEL etc. in the case of future projection) to ensure that the report change is actual change and not propagated model error. This point is the major sticking point in this article and would need to be addressed by the authors in a detailed analytical way. Such resources which conflict with the outcomes of this study are available below (to name a few):

1) TOWARDS THE SUSTAINABLE USE OF AFRICAN WETLANDS – John Pascal Simaika, Anne Alje Van Dam and Albert Chakona (eds). *Frontiers in Environmental Science*; <https://www.frontiersin.org/research-topics/9466/towards-the-sustainable-use-of-african-wetlands>

2) <https://gga.org/protecting-africas-wetlands-a-critical-component-of-environmental-governance/>

3) <https://www.wetlands.org/new-report-degradation-wetlands-sahel-wrecking-lives-driving-migration/>

4) <https://link.springer.com/article/10.1007/s00027-012-0259-2>

5) [https://www.tandfonline.com/doi/full/10.1080/00207233.2018.1494931?](https://www.tandfonline.com/doi/full/10.1080/00207233.2018.1494931?casa_token=ZOeqNdTI_d0AAAAA%3A3oj_snm_Yg52Wc8dlbTC63YDI-V1vZKUFcxGlwe2d8qfzkyPM7ZBOV3YrUf3FR0-EvdBBVxJqROOQ)

[casa_token=ZOeqNdTI_d0AAAAA%3A3oj_snm_Yg52Wc8dlbTC63YDI-V1vZKUFcxGlwe2d8qfzkyPM7ZBOV3YrUf3FR0-EvdBBVxJqROOQ](https://www.tandfonline.com/doi/full/10.1080/00207233.2018.1494931?casa_token=ZOeqNdTI_d0AAAAA%3A3oj_snm_Yg52Wc8dlbTC63YDI-V1vZKUFcxGlwe2d8qfzkyPM7ZBOV3YrUf3FR0-EvdBBVxJqROOQ)

- The introduction is missing a section on the details of threats to wetlands which could result in a wetland loss as well as the other side of the coin concerning wetland gain (what conditions/factors increase wetland distribution?)
- The HHI index needs to be expanded upon – what parameters drive it and what remote sensing variables were used to create the index? What does the scale mean when interpreting the index? Not clear
- Pg8; lines 5-7: Has population increased along the coast more so than internally (to warrant wetland loss)? You need to additional contextual support from the literature here.
- Future wetland trajectory: it would be great to breakdown the influencing parameter from these climate models to support this trend (e.g. is an increase in rainfall/SM/Temperature expected?). Also how did this section of analysis take into account the anthropogenic aspect?
- Pg12; lines 9-11: Is an increase in temperature and increase in CO2 really support an increase in wetland coverage? What about future precipitation and SM trends? Greater digging and literature support are required here.
- Pg 17: Were palustrine or vegetated wetlands (with no surface water present) included in the analysis?
- Pg18: What was your source of an independent test dataset? How was the model calibration and validation dataset collected and from what source? Later in the methods section, you mention ‘visual inspection’ – how do you maintain impartiality and reduce sampling bias?

(Remarks on code availability)

The code is clear and easy to replicate in the Google Earth Engine interface. The incorporation of an independent validation dataset from a variety of local sources would have helped improve the validity of this scripted outputs immensely.

Reviewer #3

(Remarks to the Author)

Review on “African inland wetland area on the rise 1 during the 21st century”, submitted to Nature Communications, by Li et al

African wetlands form an integral part of the continent’s ecology, so this study is making a valuable contribution to the understanding of the present and future dynamics of African wetlands. However, since climate change is a consideration here, more of the current literature for Pan African needs to be included. The work presented supports the conclusions and claims, but there may be additional important work required to justify the data interpretation and conclusions (significance of trends). Notwithstanding, enough detail is provided on the methods in order to enable reproduction of the work.

Major concerns

1. Pan African climate change work needs to be included:

- a. Engelbrecht F.A., Adegoke J., Bopape M-J., Naidoo M., Garland R., Thatcher M., McGregor J., Katzfey J., Werner M., Ichoku C. and Gatebe C. (2015). Projections of rapidly rising surface temperatures over Africa under low mitigation. *Env. Res. Letters*. 10 085004. <https://doi.org/10.1088/1748-9326/10/8/085004>.
- b. Dosio A., Jury M.W., Almazroui M., Ashfaq M., Diallo I., Engelbrecht F.A., Klutse N.A.B., Lennard C., Pinto I., Sylla M.B. and Tamoffo A.T. (2021). Projected future daily characteristics of African precipitation based on global (CMIP5, CMIP6) and regional (CORDEX, CORDEX-CORE) climate models. *Climate Dynamics* 57 3135–3158. <https://doi.org/10.1007/s00382-021-05859-w>.
- c. Dosio A, Lennard C, Spinoni J. Projections of indices of daily temperature and precipitation based on bias-adjusted CORDEX-Africa regional climate model simulations (2022). *Climatic Change* 70 13.

2. Page 4, line 3: Yes, but for some areas their small-scale losses may have huge ecological impacts. These impacts may be more visible if the study area is made smaller, e.g., only looking at the southern African region as opposed to the whole Africa.

3. Page 6, line 12: The decrease seems small over southern Africa, but is the ~2% loss from a Pan African perspective or from a southern African one?

4. A number of figures in the main text and also the supplementary material indicates linear trends, but without showing what the statistical significance levels of these trends are, the trend results are irrelevant.

(Remarks on code availability)

Version 1:

Reviewer comments:

Reviewer #2

(Remarks to the Author)

Through a very well researched rebuttal, all major comments have been addressed and the context and/or the way in which the overall results were presented has been improved with extra nuances.

Reviewer #4

(Remarks to the Author)

This is an interesting and ambitious paper that seeks to map trends in wetland extent over the period 1984-2021, and extrapolate forward to assess how climate change will impact wetlands in the future. The use of Landsat data over such a large region is impressive and the results are of general interest to a wider audience, making it potentially publishable in Nature Communications. However I have a number of reservations about aspects of the work which in my opinion need to be addressed for the work to be published here.

Major comments

1. Evaluation of interannual variability against independent datasets

Figure 1b shows the change in African wetlands from 1984 to 2021. I am somewhat familiar with the Sudd wetland and was surprised to see areas of loss and gain in close proximity to each other. Hydrologically this seems rather curious, and in contrast to a very impressive study focused on this area by Hardy et al (2023). They document interannual variability in inundation over this period also using Landsat data. They demonstrate a remarkable expansion of the Sudd, particularly since 2019, and evaluate it against independent datasets. The Sudd is one of the largest wetlands in Africa, and the apparent misdiagnosis of the long-term trend here (with areas of loss and gain) raises serious concerns for me about the quality of the mapping. I would be reassured if the authors looked carefully at that study and evaluated their own results against that benchmark.

In a similar vein, I would like to see a comparison of the interannual variability presented in this study with the GIEMS dataset of wetlands (Bernard et al, 2024) across Africa. That dataset is at much coarser spatial resolution, bringing with it its own issues (e.g. missing small wetlands), but its interannual variability at the scale of such major wetlands should be credible. As it covers the period 1992-2020, it offers an independent dataset to evaluate the current methodology.

2. Climate change, climate variability and the statistics of the rainfall-wetland relationship

The authors appear to have a view of African climate variability and change that emphasises the latter over the former. For example (P7 L1), they describe the time series of inland wetlands as “a strange trajectory”. In the context of strong interannual variability in climate (rainfall), particularly at the scale of large river catchments (e.g. Niger, Congo, Nile), I don’t

see this variability as strange at all. I would expect strong interannual variability in wetland extent dominating over any continental-scale rainfall trends. Instead (P1 L23) they state that “change of wetland area [is] closely related to climate change.”

The authors state (P10 L7-8) that “from 1984-2006... precipitation decreased significantly and continuously”, based on 4 time points. What accumulation period do these 4 time points actually represent? To justify their statement about “significant and continuous”, they should show interannual rainfall variability and test the rainfall trend over the period 1984-2006. The authors should also note the high degree of uncertainty in African precipitation trends across datasets (Maidment et al, 2015). The precipitation in the dataset that they use (ERA5) is not considered particularly reliable. In this context, a much better choice would be the ERA5-Land dataset (<https://www.ecmwf.int/en/forecasts/dataset/ecmwf-reanalysis-v5-land>), where they correct biases in the raw ERA5 precipitation.

Related to this, the authors state (P10 L5-9) that “the change in precipitation explains the violent fluctuation of wetland time series”, based on Figure 3b. The authors need to quantify how much of the variability the precipitation time series actually explains with a linear regression (and associated P-value). Given the lack of time points in their dataset, I would strongly recommend that they also perform a series of correlation analyses at sub-continental scale. Interannual rainfall variability is not spatially coherent at the continental scale, and consistent relationships between smaller-scale droughts and pluvials and wetland extent over effectively independent regions would greatly strengthen their arguments. Of course, it is reasonable to expect strong interannual rainfall variability to drive wetland extent, but Figure 3b does a poor job of evidencing that relationship.

3. Cloud contamination

Apart from the brief mention in SI (P4) about the omission of the 1991-1995 period, there is no discussion of the problem of estimating wetland extent using Landsat in cloud-dominated regions (most notably the Congo Basin). At the very least, I would expect the paucity of cloud-free images in such areas would enhance the uncertainty in the resulting wetland mapping there. Is that not the case? I would like to see some evidence of that.

4. Representation of fluvially-driven inundation

Clarification is needed for how TOPMODEL represents wetlands which are primarily driven by non-local rainfall. For example, interannual fluctuations in the Sudd and Niger Inland Delta are dominated by rain that fell hundreds of kilometres away and that spills out on the floodplain. I found it hard to infer how large a region contributes to the precipitation used in these areas from the description in SI section 8, which mentions over 40,000 watershed units. Would a map showing (at least the larger) watershed units help, or are they all too small to be represented on an African map? If the latter is the case, what does it mean to assume that TOPMODEL can simulate wetlands such as the Sudd based only on local rainfall? I assume that assimilation of remotely-sensed soil moisture data in FLDAS implicitly captures fluvial-driven variability in such cases, but that source of information is not available in the climate change simulations, which do not depict the key hydrological processes.

Minor points

P3 L2 and throughout: The authors refer to “10 periods from 1984 to 2021” though only 9 points are shown in Figures 2 and 3. In SI (P4) they note that the period 1991-5 suffered from lack of data and presumably this explains its absence in those figures. For consistency in the main text, it would make sense to talk about analysis of 9 periods, not 10.

The authors overstate the case with respect to the historical trend in inland wetland (P6 L19) “completely opposite trends... in coastal and inland wetlands”. Whilst the downward trend in coastal wetland is statistically robust, the inland trend ($P=0.175$) cannot be distinguished from zero at even the 10% level.

P8 L10 “has been conversion to...”

I may have missed it, but what soil moisture levels (in FLDAS and the CMIP6 models) were actually used? On a related note, the y-axis range in SI Fig. 20 (~650 m³/m³) is clearly wrong.

SI Figure 19 shows that TOPMODEL forced by FLDAS soil moisture does a good job at representing interannual wetland variability. Can the authors clarify how the calibration procedure (to determine the value of the M parameter) has influenced this simulation? In other words, was there data held back from the calibration step to evaluate the model simulations?

In Figure 5, what range of values does a “constant” wetland area correspond to in the bar charts? In the maps, it would be better to depict that “constant” orange over a range at equal distance either side of zero (e.g. -3 to +3).

References

- Bernard, J., Prigent, C., Jimenez, C., Fluet-Chouinard, E., Lehner, B., Salmon, E., Ciais, P., Zhang, Z., Peng, S., & Saunois, M. (2024). The GIEMS-MethaneCentric database: a dynamic and comprehensive global product of methane-emitting aquatic areas. *Earth Syst. Sci. Data Discuss.*, 2024, 1-35. <https://doi.org/10.5194/essd-2024-466>
- Hardy, A., Palmer, P. I., & Oakes, G. (2023). Satellite data reveal how Sudd wetland dynamics are linked with globally-significant methane emissions. *Environmental Research Letters*, 18(7), 074044. <https://doi.org/10.1088/1748-9326/ace272>
- Maidment, R. I., Allan, R. P., & Black, E. (2015). Recent observed and simulated changes in precipitation over Africa. *Geophysical Research Letters*, 42(19), 8155-8164. <https://doi.org/10.1002/2015gl065765>

Version 2:

Reviewer comments:

Reviewer #4

(Remarks to the Author)

The authors have provided a lot of material in response to my comments! Thank you for that considerable effort. Below I flag some areas where I am still unclear about their logic. On several points, the authors do not appear to have made any change to the manuscript. I would like to see updates in the manuscript consistent with evidence presented in their response. It may be that I missed some changes in the manuscript, but lines featuring yellow highlighting in the revised manuscript are remarkably few given the 35-page response.

Major point 1

The authors have done a good job of comparing their mapping with the TropWet dataset of Hardy et al, and arguing why their analysis differs from that study. I also appreciate their efforts to provide additional evaluation, particularly with respect to the GIEMS dataset. However, there are several points I must query them on. In their response they state (p5) "As illustrated, the interannual trajectory of African wetland area in GIEMS-MC closely mirrors our results: an initial decline followed by a subsequent increase." I agree that "the subsequent increase" in GIEMS is very clear in their response Figure 3. However, it is quite misleading to describe the earlier part of the GIEMS time series as "an initial decline". I struggle to see how the comparison "partially validates our pre-2000 findings". I believe the authors when they argue that the limited availability of Landsat and other data pre-2000 affects the quality of the mapping. I think the authors need to add a sentence in the main text. I also think that the comparison they show in the response Figure 3 should be noted in the text and included in SI, though given the strong interannual variability in the GIEMS dataset, I don't really see the value of the polynomial fits they present. I am also somewhat surprised that the authors don't provide a correlation metric between the two datasets as, by eye, the comparison seems to provide important independent evaluation of their findings.

Major point 2

Firstly, I must apologise for having provided some misleading information in my original review. As I have subsequently learned, precipitation in ERA5-Land is not in fact bias-corrected with observations. In that context, the similarity between ERA5_Land and ERA5 in the response Figure 7 is not surprising. Their response Figure 8 is rather revealing of the general problem that I raised on the other hand, with ERA5-Land showing a quite distinct negative trend prior to 2000 compared to the other datasets. Having gone to some effort to consider 3 other datasets, I am puzzled why they have not updated the main text consistently. On p10 (L7-8) they state that "From 1984-2006 there is a sustained decline in precipitation". This statement is apparently based on the FLDAS dataset yet it looks at odd with FLDAS in response figure 8. Are the points in Figure 3 based on single year values from the response figure 8? That would be curious given the large interannual variability evident in response figure 8. Also the inconsistency in the 1987 value of soil moisture (and PDSI) compared to precipitation makes me wonder if they have plotted ERA5 precipitation data in Figure 3. In any case, response figure 8 provides no support for the apparent "sustained decline" if one ignores (or perhaps even weights equally) ERA5-Land. The authors have done a lot of welcome additional regional analysis (their "issue 2") of precipitation and wetland area datasets in response tables 1-3 and associated response figures. I am curious though about some of the very low p-values reported given the small number of data points (n=9 for wetland area trends). I think the regional analysis provides important supporting evidence that wetland area is responding to changes in precipitation. However, in the revised manuscript the authors have not adequately responded to my comments about "climate change". The vast majority of readers will interpret the statement in the abstract "the change of inland wetland area is closely related to climate change" as meaning long-term anthropogenic climate change, a message reinforced by the phrase in the title "on the rise during the 21st century". There is still no reference in the text to the importance of regional-scale decadal variability in precipitation.

Major point 3

The authors provide a very lengthy discussion of issues related to cloud contamination in their response. This concludes with a statement that "the data from post-2000 are essentially unaffected by clouds". Did I miss the evidence for this, or is it simply an assertion? I suggest that the earlier comparison with GIEMS data (unaffected by cloud) does provide evidence, but they don't mention it here. There is still no mention of cloud issues in the text as far as I can see.

Major point 4

The authors have responded to most of my comments, but I am still unclear if the lack of fluvially-driven information implicit in FLDAS within the climate change simulations means that here is an important missing hydrological processes in some catchments.

Version 3:

Reviewer comments:

Reviewer #4

(Remarks to the Author)

I'd like to thank the authors for responding fully to my original comments and incorporating elements of the response into the main text, methods and SI. I think they have successfully done that without disrupting the logical flow and need for brevity in a Nature Communications article. I recommend the manuscript for publication.

For information, the answer to the question raised by the authors in response to my "major point 4" (below) is yes.

"Is the reviewer referring to whether, in the future climate simulations, the absence of FLDAS (soil moisture reanalysis assimilation data) means that the model omits fluvially-driven hydrological processes?"

Reviewer#2

General Comments: The overall observation of a rise in wetland coverage in the future and the downplaying of wetland loss in the past goes against a lot of independent reports and academic articles. It is well-documented that population explosions and the increasingly drying of the African continent will have dire consequences on the environment and wetlands. The authors need to ensure rigorous error analysis (i.e. error propagation and how it influences the rate of change over the time series before future projection; and the error linked to the TOPMODEL etc. in the case of future projection) to ensure that the report change is actual change and not propagated model error. This point is the major sticking point in this article and would need to be addressed by the authors in a detailed analytical way. Such resources which conflict with the outcomes of this study are available below (to name a few).

1) Response: We would like to express our gratitude to the reviewer for his/her suggested comments. The feedback and critiques not only helped us identify shortcomings in our manuscript but also provided a clear guidance for an further improvement in our research work.

The reviewer mentioned that the conclusions of our writing contradict many independent reports and academic articles, so we would like to clarify this point further. The statement in our manuscript that "there was no large-scale reduction in African wetlands from 1984 to 2021" may have been somewhat misleading. We acknowledge this miscommunication in our wording, and we have made some revisions to the conclusion section of the main text to address this issue. The revised details can be found on [Page 17, lines 03-05]:

Our findings support the following conclusions: (1) There was no large-scale loss of African wetlands 1984 to 2021. The losses were concentrated in the coastal areas, the Congo Basin and southern Africa.

[Page 17, lines 12-14]:

An in-depth understanding of wetlands is fundamental to their conservation and utilization. Even in the context of rising wetland area, irrational human activities (e.g., damming, reclamation, and irrigation) can still bring humanitarian disasters to residents, and the focused conservation of Africa's wetlands remain urgent and necessary. We hope that our work will contribute to improving the management and conservation of African ecosystems and, in the process, help support those who depend on these ecosystems.

[Page 16, lines 10-16]:

It is important to note that although Africa's wetlands have not experienced significant overall loss, there has been substantial reduction in certain regions (e.g., the Congo Basin, Southern Africa) and in coastal wetlands. The quality of Africa's wetlands still needs to be assessed, as stable or expanding wetland areas do not necessarily indicate a favorable condition. The degradation of the ecological functions of existing wetlands continues to pose serious threats to local communities, for example, the shrinking of Lake Chad's surface area.

In this study, we classified African wetlands into two major categories—inland wetlands and coastal wetlands. The trend and trajectory of coastal wetlands showed a significant decline, a conclusion that aligns with the mainstream research on coastal wetlands, including the Nature paper "The global distribution and trajectory of tidal flats" and the Science paper "High-resolution mapping of losses and gains of Earth's tidal wetlands. "

Regarding African inland wetlands, our research results indicate a trajectory of initial decline followed by a subsequent increase. Specifically, from 1984 to around 2000, the area of African inland wetlands exhibited a downward trend. However, after 2000, the area began to increase, which is reflected in our statement that "African inland wetland area on the rise during the 21st century."

We are very grateful to the reviewer for providing numerous references and news reports related to the loss and degradation of African wetlands. However, it is important to note that most of the existing independent reports and academic articles on Africa are limited to localized field surveys and lack comprehensive studies on the African

continent as a whole. While our results do reflect the wetland losses mentioned in some of these reports, there is no fundamental conflict with their conclusions or statements. Notably, many of the wetland losses identified in these field reports occurred before 2000, which is consistent with our study's findings.

We addressed the reports and papers cited by the reviewer as examples.

1) Editorial: Towards the Sustainable Use of African Wetlands

This article underscored the importance and necessity of sustainable development of African wetlands, and we fully agreed with the viewpoints expressed. The article began by highlighting a characteristic of African wetland research, namely the "limited availability of data." Davidson's paper particularly emphasized the "need to enhance awareness of wetland changes globally, especially in regions like Africa," and noted that "the evidence of wetland area changes published was incomplete and limited, particularly for Africa, Neotropics, and Oceania."

The descriptions of wetland area loss in this editorial mainly stemmed from assessments using the WET Index¹. However, this made the paper's conclusions about losses in Africa still unreliable. The WET Index was an indicator of wetland area changes based on incomplete and heterogeneous data (i.e., time series records of wetland extents from published literature), and it was not a direct observation of African wetlands. While the WET Index could reflect global wetland range change trends to some extent, it also had considerable limitations. This was particularly true for Africa, where the construction of the WET relied heavily on unevenly distributed wetland records concentrated in a few countries such as Uganda, Kenya, and in North African nations like Algeria and Libya, as shown in the figure. It should have been acknowledged that the accuracy of this index in predicting wetland changes in Africa was significantly lower than in regions like Europe, the USA, and China. We believed that estimates of wetland change trends in Africa based on the WET Index should have been treated with considerable uncertainty and were likely less reliable than our results obtained from remote sensing classification.

[Figure Redacted]

Figure 1. Time series data used to construct the WET Index (from the paper "Improvements to the Wetland Extent Trends (WET) index as a tool for monitoring natural and human-made wetlands").

2) Protecting Africa's wetlands: critical to environmental governance

This article highlighted that wetlands are crucial resources, and protecting African wetlands was vital for environmental governance. The article discussed the state of African wetlands, particularly focusing on the degradation and disappearance of wetlands in South Africa and Zimbabwe. We fully agreed with the issues identified in the article, as our research also yielded similar results. As depicted in Figure 1b of the main text, not only in South Africa and Zimbabwe but across the entire southern part of Africa (regions south of 10° South latitude), a significant degree of wetland loss was evident.

Since the wetland area in this region does not constitute a large percentage of the continent's total wetland area, we did not focus on this area in our paper, which was a significant oversight in our wetland research. We have already made certain additions to the main text to address this. The revised details can be found on [Page 06, lines 15-17]:

In addition, the entire southern part of Africa (south of 10°S latitude) showed

considerable wetland loss and little wetland growth compared to other parts of Africa.

Figure 2. Wetlands in Southern Africa have experienced area loss.

3) New report: Degradation of wetlands in the Sahel is wrecking lives, driving migration

This article discussed a report titled "Water Shock: Wetlands and Human Migration in the Sahel Region," which explored how poor management of water resources led to ecosystem degradation and triggered human migration, including relocation to Europe.

The report highlighted Lake Chad as one of the most well-known examples of wetland degradation in Africa. The article stated that "due to water extraction for irrigation projects, the Lake Chad basin had lost 95% of its surface water area," a

statement we acknowledged as accurate. However, it was essential to note that the degradation of Lake Chad referred to in the report was solely the lake surface, that is, a reduction in the surface water area. Observations of the current state of Lake Chad revealed that the degraded water surface had not been completely exposed as dry land but was covered by herbaceous vegetation. The region's soil maintained waterlogged conditions or seasonal water accumulation, considered a transitional zone between aquatic and terrestrial ecosystems. In our research definition, it was still considered part of the wetlands.

[Figure Redacted]

Figure 3. Degradation of the Lake Chad wetlands, showing the shrinkage of the lake surface.

The phenomenon of wetland degradation should have indeed raised concern. However, this did not conflict with our research findings. In our study, such flooded grasslands were also considered part of herbaceous marshes. Therefore, when calculating the area of wetlands in this region, we believed it was only appropriate to emphasize the degradation of the water surface, rather than a reduction or disappearance of the entire wetland area. Of course, we also added and emphasized this issue in the main text, noting that the stability or expansion of wetland area did not necessarily

indicate the health of the wetlands. The ongoing degradation of existing wetlands could still pose a severe crisis to local residents (e.g., Lake Chad).

The additional content can be found on [Page 16, lines 10-16]:

It is important to note that although Africa's wetlands have not experienced significant overall loss, there has been substantial reduction in certain regions (e.g., the Congo Basin, Southern Africa) and in coastal wetlands. The quality of Africa's wetlands still needs to be assessed, as stable or expanding wetland areas do not necessarily indicate a favorable condition. The degradation of the ecological functions of existing wetlands continues to pose serious threats to local communities, for example, the shrinking of Lake Chad's surface area.

4) The status of wetlands, threats and the predicted effect of global climate change: the situation in Sub-Saharan Africa

This paper reviewed the major threats faced by wetlands in Sub-Saharan Africa through case studies, including typical wetland areas such as the Niger River, Lake Chad, the Sudd Wetlands, and the Congo Basin. The paper discussed various human activities (such as water storage, irrigation, hydroelectric power generation) and climate change threats to African wetlands.

Losses in wetlands like the Niger Delta, Lake Chad, and the Congo Basin, mentioned in the paper, were all reflected in Figure 1b of our paper. Our advantage was that we conducted a comprehensive analysis from a remote sensing perspective, which included not only typical regions like the Niger River, Lake Chad, or the Congo Basin but also various small lake wetlands. This comprehensive analysis led to the conclusion that African wetland areas, on the whole, were increasing. Additionally, the paper heavily cited data from field surveys and analyzed the impacts of population growth and changes in local laws and policies on the wetlands of the region, including impacts on fish, migratory birds, and other wildlife. These survey data were mostly concentrated before the year 2000 (mid-20th century), which is consistent with our research findings that the area of African wetlands showed a declining trend from 1984 to around 2000.

[Figure Redacted]

Figure 4. Loss of Wetlands in Africa.

Regarding climate change predictions, the paper was published in 2013 and primarily based its climate change assessment on the IPCC Fourth Assessment Report (2007). However, the latest CMIP6 simulations now offer different forecast results. Research shows that, apart from Southern Africa and Madagascar, most regions of Africa are expected to experience a fivefold increase in extremely wet years, particularly in the Sahel region, where the annual average precipitation is projected to

increase^{2,3}. For a detailed discussion, refer to the supplementary material—Climate Change in Africa.

5) A spatial and temporal assessment of wetland loss to development projects the case of the Kampala – Mukono Corridor wetlands in Uganda

The paper reported on the assessment of spatial and temporal losses of wetlands in the Kampala-Mukono Corridor (KMC) due to development projects (DP). The discussion in this paper is similar to previous reports, and we acknowledge the wetland losses caused by human activities (in this case, project development). However, since the focus of our study was not on some specific areas in Africa with severe losses, these details were not mentioned in our manuscript.

2) Response: We fully understood the reviewer's concerns about distortions in results caused by errors. These concerns mainly include two aspects: the rate of change in the time series of wetland area during historical periods and the impact of uncertainties in historical time series on the future prediction. We have made targeted modifications, additions, and explanations to address these issues.

The reviewer pointed out the lack of a rigorous error analysis. In the manuscript, we validated the effectiveness of our classification results using remote sensing classification confusion matrices, comparisons with the other wetland maps, and comparisons with TOPMODEL simulation results. The error range for the classification of African wetlands was not mentioned in the original paper, but has been addressed in this revision.

Following the methods used in the Science journal article "High-resolution mapping of losses and gains of Earth's tidal wetlands," we employed the bootstrap resampling method to calculate the 95% confidence intervals for wetland classification accuracy and area across ten historical periods (For specific steps, see Supplementary Discussion 5 - Uncertainty of African Wetland Classification).

The mean of the sampling distribution is used as the estimate of accuracy for the study, and the 0.025 and 0.975 percentiles of the sampling distribution are used as the 95% confidence intervals. The accuracy and area results are presented in the following

table:

Table 1. Confidence Intervals for Wetland Classification Accuracy

Year	OA	OA_Lower	OA_Upper	UA	UA_Lower	UA_Upper	PA	PA_Lower	PA_Upper
1987	0.957	0.945	0.967	0.941	0.903	0.975	0.753	0.689	0.810
1993	0.959	0.948	0.969	0.904	0.860	0.943	0.771	0.708	0.827
1998	0.964	0.954	0.973	0.932	0.895	0.963	0.827	0.776	0.874
2002	0.970	0.961	0.978	0.920	0.882	0.953	0.876	0.833	0.915
2005	0.959	0.949	0.969	0.877	0.827	0.925	0.761	0.697	0.826
2008	0.970	0.961	0.979	0.948	0.917	0.975	0.855	0.811	0.900
2011	0.967	0.957	0.976	0.923	0.882	0.958	0.843	0.789	0.889
2014	0.969	0.960	0.977	0.902	0.862	0.941	0.854	0.805	0.901
2017	0.968	0.959	0.977	0.868	0.819	0.913	0.862	0.807	0.909
2020	0.967	0.958	0.976	0.926	0.885	0.961	0.837	0.787	0.883

OA: Overall Accuracy; UA: User Accuracy; PA: Producer Accuracy

Table 2. Confidence Intervals for Wetland Classification Area (km²)

Year	AW	AW_Lower	AW_Upper	CW	CW_Lower	CW_Upper	IW	IW_Lower	IW_Upper
1987	1200819.68	1500699.641	1100116.463	119543.48	141726.8062	109275.9418	1081276.2	1376995.219	983871.3555
1998	1184975.43	1458040.776	1081358.445	113697.99	134029.2562	106336.2496	1071277.44	1336335.776	966274.3558
2002	1173041.54	1447062.656	1065721.009	109747.14	128857.3385	99161.59913	1063294.4	1339030.695	960445.7763
2005	1163539.39	1437120.484	1039141.793	112893.96	131814.9177	103377.7366	1050645.43	1326521.879	923863.2727
2008	1185585.18	1444905.975	1077920.931	113547.1	144072.8619	103611.7288	1072038.08	1307338.387	966963.8107
2011	1188910.32	1430237.024	1076535.806	109208.78	131866.0773	102589.1519	1079701.54	1309841.039	960819.6479
2014	1193304.48	1394771.47	1075370.338	107757.58	122707.4735	102300.3441	1085546.9	1290954.941	958681.4366
2017	1180854.93	1356023.484	1068016.764	107551.51	125476.7617	102804.8331	1073303.42	1237423.243	953582.0155
2020	1194688.76	1381294.592	1092780.026	108015.11	127957.2642	101347.5106	1086673.65	1261103.118	981261.077

AW: African Wetlands; CW: Coastal Wetlands; IW: Inland Wetlands

Regarding the impact of uncertainties in historical time series data on future wetland trend predictions, we consider it to be negligible.

Our research utilized the TOPMODEL approach, which is mainly based on the Nature Climate Change journal paper “Future impacts of climate change on inland Ramsar wetlands.” This paper highlighted that uncertainties in simulation results primarily arise from three sources: the input soil water data, the choice of wetland calibration data and methods, and the inherent uncertainties of the TOPMODEL itself. The reviewer's main concern pertains to the second source of uncertainty, specifically the use of historical period classification results (which include uncertain wetland area

time series) as calibration data, potentially impacting the reliability of future wetland predictions.

It's important to clarify that the use of wetland data for calibration in TOPMODEL is solely for adjusting the simulated wetland area size and does not involve correcting interannual changes or trends. The parameter that requires calibration in the model's formula is M , which is an adjustable parameter describing the exponential decrease in soil permeability with depth. The M value varies across different basins, and its range is set between 1 and 15. The M value derived from wetland classification results is a fixed value for a basin and does not change with future simulation time. Therefore, we believe that the impact of historical period time series uncertainties on future predictions is limited to the relative size of wetland areas, and in principle, it does not affect interannual changes and trends.

The uncertainty in the input soil water data is a major factor affecting future wetland simulation results. In this study, we used soil moisture data from 14 CMIP6 models as input, with uncertainties mainly reflected in the significant differences between these model data. These differences are far greater than the uncertainties introduced by model error propagation. We have provided different time series curves (Figure 4 in the main text) for these input data and analyzed them. Additionally, the inherent uncertainty of the TOPMODEL itself is demonstrated in the supplementary material. For specific information, please refer to Supplementary discussions 6—Effectiveness and Uncertainty of Inland Wetland Simulation.

3) Response: Regarding the statement that "population explosion and increasing aridity across the African continent will have severe consequences for the environment and wetlands," we can only agree in part. Indeed, the dramatic increase in Africa's population has exerted significant pressure and impact on local environments and ecosystems, including wetland ecosystems. In our manuscript, we conducted a preliminary assessment and analysis using the Human Influence Index. However, regarding the future climate change conditions on the African continent, there is no consensus among existing studies. The assertion that the African continent is becoming

increasingly arid is not reliable.

According to the latest CMIP6 simulation results, except for Southern Africa and Madagascar, most regions of Africa are expected to experience a fivefold increase in extremely wet years, particularly in the Sahel region, where the annual average precipitation is projected to increase. The paper published in Nature Ecology & Evolution, "Human population growth offsets climate-driven increase in woody vegetation in sub-Saharan Africa," mentions that due to increases in carbon dioxide and precipitation, even arid lands in Africa are showing a greening trend. In the Sahel region, future precipitation is not predicted to decrease. Forecasts indicate that the average rainfall in the area will remain relatively stable, but the frequency of droughts and extreme rainfall events may increase⁴.

[Figure Redacted]

Figure 5. Future Precipitation Changes in the Sahel Region (from the paper "A Global Perspective on African Climate").

Additionally, multiple studies have indicated that the global wetland area is expected to increase over the next century (21st century). For instance:

1. Zhang et al. used the LPJwsl land surface process model (incorporating TOPMODEL for simulating wetlands) to simulate the dynamics of global inland wetlands in the 21st century under four climate warming scenarios of the Coupled Model Intercomparison Project Phase 5 (CMIP5). They found that by the end of the 21st century, the area of global inland wetlands is projected to increase by 5%, 8%, 12%, and 20% under the four scenarios, respectively. The increase in wetland area is mainly concentrated in the high latitudes of the Northern Hemisphere, while tropical

wetland areas slightly decreased⁵.

2. Comyn-Platt et al., utilizing output from a medium-complexity climate model to drive the land surface process model JULES (which uses TOPMODEL for simulating wetlands), discovered that wetland areas in permafrost regions are expected to increase by 0.26 million km² and 0.28 million km² under future warming scenarios of 1.5° C and 2° C, respectively⁶.

3. Kleinen and others used the MPI-ESM Earth system model to simulate changes in global wetlands from 1860 to 3000. They predict that tropical wetlands will expand due to increased precipitation over the next 1000 years, while high-latitude wetlands will expand due to a reduction in ice cover⁷.

In summary, the results of our study do not conflict with existing mainstream research on African wetlands. Our research focused on the overall trend of changes in African wetlands. Although we also observed localized losses of wetlands, we did not emphasize these findings, thereby overlooking the pressures and challenges that localized wetland degradation poses to local residents in Africa. This oversight is a limitation of our paper.

Prompted by the reviewer's comments, we have recognized that emphasizing the overall increase in African wetlands could potentially mislead policy making and conservation efforts. Consequently, we have added content to our paper to further stress that, even against a backdrop of increasing wetland areas, unsustainable human activities (e.g., damming, water storage, and irrigation) can still lead to humanitarian disasters for local communities. Thus, the focused conservation of African wetlands remains urgent and necessary.

1.Comment: The introduction is missing a section on the details of threats to wetlands which could result in a wetland loss as well as the other side of the coin concerning wetland gain (what conditions/factors increase wetland distribution?)

Response: This study primarily utilized historical wetland classification and future

period simulation methods to reveal the overall trend of wetland area changes in Africa during the 21st century. The manuscript does not delve deeply into the driving factors behind these changes, whether they involve loss or increase of wetlands.

We recognized that due to Africa's vast geographical and cultural diversity, the factors contributing to wetland loss and gain vary significantly across different regions. Our research has merely analyzed the overall changes in African wetlands. A more detailed and in-depth study, such as examining the specific impacts of human activities (population growth, water usage, agricultural expansion, etc.) on wetlands, will be our next research focus.

Indeed, incorporating details about threats to wetlands in studies of spatial and temporal changes in African wetlands is essential, as this can draw attention to the need for conservation of African wetlands. Therefore, we have added the following content at the end of the paper. The revised details can be found on [Page 17, lines 12-14]:

An in-depth understanding of wetlands is fundamental to their conservation and utilization. Even in the context of rising wetland area, irrational human activities (e.g., damming, reclamation, and irrigation) can still bring humanitarian disasters to local residents, and focused conservation of Africa's wetlands remains urgent and necessary. We hope that our work will contribute to improving the management and conservation of African ecosystems and, in the process, help support those who depend on these ecosystems.

2.Comment: The HHI index needs to be expanded upon – what parameters drive it and what remote sensing variables were used to create the index? What does the scale mean when interpreting the index? Not clear

Response: We apologize for any confusion caused by the inappropriate wording in our manuscript. In this study, we utilized the Human Influence Index (HII) to assess the impact of human activities on African coastal and inland wetlands. The HII is a composite indicator that quantifies human pressure on the environment. This pressure, often referred to as a threat to biodiversity, arises from human actions that may harm nature.

The HII is composed of nine global data layers, encompassing human population pressure (population density), human land use and infrastructure (built areas, night-time lights, land use/land cover), and human access routes (coastlines, roads, railways, navigable rivers).

This means that a higher HII value for a region indicates greater impact from factors such as population growth, land development, or various transportation routes. In our research, we used the HII as a proxy for the direct and indirect impacts of various human activities (such as water storage, irrigation, hydroelectric power) on African wetlands. The advantage of using HII is that it allows for a consistent temporal analysis with HII maps, providing a comprehensive overview of the various human pressures on African wetlands, especially those low-intensity pressures that are difficult to detect via remote sensing.

3.Comment: Pg8; lines 5-7: Has population increased along the coast more so than internally (to warrant wetland loss)? You need to additional contextual support from the literature here

Response: Thank you very much for this helpful suggestion.

In the text, we stated, "Population growth and increasing economic development have caused some degree of destruction to coastal wetlands around the world." This conclusion is derived from the Millennium Ecosystem Assessment report "Ecosystems and Human Well-being: Wetlands and Water," which identifies population growth and increasing economic development as major indirect drivers of degradation and loss of both inland and coastal wetlands.

The reviewer may have misunderstood the intent of this statement. We did not mean to suggest that population growth in African coastal areas necessarily exceeds that in inland areas. As previously mentioned, our analysis of drivers did not delve into any specific human activity factors (for example, population growth, land reclamation, and construction of water management facilities) but instead used the HII as a composite index to assess human pressures on African wetlands.

Based on our findings, the HII values in coastal areas are more than twice those in

inland areas, indicating that coastal wetlands face significantly higher human pressures than inland wetlands, which substantiates the rationale behind coastal wetland loss. Regarding the differences between coastal and inland areas in terms of population and urban development and their impact on African wetlands, we plan to conduct a more thorough discussion using case study methods in future research.

4.Comment: Future wetland trajectory: it would be great to breakdown the influencing parameter from these climate models to support this trend (e.g. is an increase in rainfall/SM/Temperature expected?). Also how did this section of analysis take into account the anthropogenic aspect?

Response: Thank you very much for this reviewer's suggestion. We have added to this content, specifically in Supplementary Discussion 2—Climate change in Africa.

Supplementary Figure. 14 | The spatial distribution of temperature and precipitation changes by 2100 under the SSP126 scenario.

Supplementary Figure. 15 | The spatial distribution of soil moisture changes by 2100 under the SSP126 scenario.

Here, we have also added spatial changes and time series graphs of temperature, precipitation, and soil moisture under four SSP scenarios, to further aid to understand the reviewer's comments.

Figure 6. Spatial distribution of future temperature changes in Africa from 2015 to 2100. Panels a-d correspond to SSP scenarios 126, 245, 370, and 585 respectively.

Figure 7. Spatial distribution of future precipitation changes in Africa from 2015 to 2100. Panels a-d correspond to SSP scenarios 126, 245, 370, and 585 respectively.

Figure 8. Spatial distribution of future soil moisture changes in Africa from 2015 to 2100. Panels a-d correspond to SSP scenarios 126, 245, 370, and 585 respectively.

Figure 9. Time series of temperature changes.

Figure 10. Time series of precipitation changes.

Figure 11. Time series of soil moisture changes.

For this part of the analysis, we did not directly incorporate human factors. However, the CMIP6 climate simulation data that we used factored in socioeconomic development scenarios (different from SSP scenarios) during their production. Therefore, it can be argued that human factors were indirectly considered to some extent in our analysis.

The Shared Socioeconomic Pathways (SSPs) are a set of new emissions scenarios driven by different socioeconomic assumptions designed to simulate changes in societal factors throughout the 21st century, including population, economic growth, education, urbanization, and the rate of technological development. For example, SSP5-8.5, characterized as a high emissions pathway, envisions a future with high fossil fuel consumption where the global economy grows rapidly, and the population increases substantially. Therefore, the CMIP6 climate data consider certain human factors during simulations. However, we have not yet conducted an in-depth study on the direct impacts of human factors on African wetlands up to the year 2100. The primary reason is that compared to climatic factors, human factors are significantly influenced by local government policies, laws, and economic development levels, which may introduce

considerable uncertainty. Currently, there are no reliable data on future human factors available for wetland simulation and analysis.

Although current results show that the area of inland wetlands in Africa has been increasing since the year 2000 and climate factor simulations also predict further expansion of inland wetlands in the future, human factors will undoubtedly have negative impacts on African wetlands going forward. This represents one source of uncertainty in future wetland simulations. We have addressed this in the section "Uncertainty of African Wetland Classification" in the supplementary materials.

5.Comment: Pg12; lines 9-11: Is an increase in temperature and increase in CO2 really support an increase in wetland coverage? What about future precipitation and SM trends? Greater digging and literature support are required here.

Response: Thank you to the reviewer for pointing out the areas of our manuscript that were not clearly articulated.

“It is found that the inland wetland area will have a net increase of more than 10 % under the SSP126/245/370/585 scenarios from 2023 to 2100, and the more intense the SSP scenario is, featuring higher global temperature rise and more carbon dioxide emissions, the higher the degree of wetland area expansion can be anticipated.”

Regarding the sentence in our paper, we intended to convey that the extent of future wetland expansion increases progressively from SSP126 to SSP585. Temperature and CO2 were mentioned merely as representations of different SSP scenarios. We apologize for any ambiguity caused by our wording, and we have made the following modifications to the text. The revised details can be found on [Page 12, lines 09-11]:

Based on the average of the results of 14 simulations, it is found that the inland wetland area will have a net increase of more than 10 % under the SSP126/245/370/585 scenarios from 2023 to 2100.

For the future trends in precipitation and soil moisture in Africa, we discussed and supplemented in the previous question.

6.Comment: Pg 17: Were palustrine or vegetated wetlands (with no surface water present) included in the analysis?

Response: This study was divided into a spatiotemporal change analysis for the historical period (1984-2021) and a wetland simulation analysis for the future period (up to 2100).

For the analysis of wetlands during the historical period, we examined the loss and gain across eight categories of wetlands, including those with vegetation. In our future wetland simulations, we also included inland wetlands or vegetated wetlands that lack surface water. However, large inland lakes such as Lake Victoria, Lake Tanganyika, and Lake Malawi, as well as coastal wetlands, were not included in the analysis.

7.Comment: Pg18: What was your source of an independent test dataset? How was the model calibration and validation dataset collected and from what source? Later in the methods section, you mention ‘visual inspection’ – how do you maintain impartiality and reduce sampling bias?

1) Response: For the validation of wetland classification results, independent test datasets primarily came from two sources:

1. Approximately 270,000 visually interpreted sampling points were deployed across various locations in Africa. Of these, 30% of the sampling points were not used for training the random forest classifier but were instead used as independent validation points for post-classification accuracy verification, such as calculating confusion matrices.

2. A certain number of wetland validation points were randomly selected from the global Regularly Flooded Wetlands (RFW), the Congo Basin forest wetlands dataset (CARPE wetlands map), and the World Mangrove Atlas, to assess the consistency of this wetland classification product.

2) Response: The calibration data for the TOPMODEL consisted of ten maps showing the distribution of African wetlands from 1984 to 2021, obtained through historical period classification.

This study did not use other wetland datasets from those such as the Global Inundation Estimate from Multiple Satellites 2 (GIEMS-2), the Global Wetland Area and Dynamics for Methane Modeling (WAD2M), or the global Regularly Flooded Wetlands (RFW). This decision was primarily because these datasets have a spatial resolution of 0.25° (at an equatorial spatial resolution of approximately 28 kilometers), which significantly reduces the accuracy of wetland area extraction compared to the results of our wetland classification, which has a 30-meter spatial resolution.

Since TOPMODEL simulates wetland distribution up to the year 2100, there are no available wetland datasets to validate the effectiveness of the simulation. In this study, we primarily validated the accuracy of historical period wetland simulations to indirectly demonstrate the effectiveness of the TOPMODEL approach. For a detailed explanation, see Supplementary Discussion 6 - Effectiveness and Uncertainty of Inland Wetland Simulation.

3) Response: Visual interpretation for obtaining training data is a common method in the field of remote sensing classification. Compared to field surveys, it is the most efficient and cost-effective method.

The training data for this study were obtained by employing three experienced remote sensing professionals, along with the author. They used high-resolution Google Earth Engine's imagery, wet and dry season Landsat images, Digital Elevation Model (DEM) imagery, and vegetation index imagery to classify wetland categories at 270,000 uniformly distributed sample points across Africa. For areas of contention, we also referred to existing wetland maps, such as the Global Regularly Flooded Wetlands (RFW) and the Congo Basin forest wetlands dataset for additional guidance in our judgments.

There is no robust method to validate the training dataset obtained through visual interpretation. Generally, validation is performed by assessing the accuracy of the final classification results.

Concerning sampling bias, we believe it is not a significant issue. To extract wetland information as accurately and comprehensively as possible, we used a

systematic uniform sampling approach based on a 0.1° x 0.1° latitude-longitude grid. The distribution and density of the sampling points were designed to closely approximate the actual state of the wetlands (see Supplementary Figure 2).

Reviewer#3

General Comments: African wetlands form an integral part of the continent's ecology, so this study is making a valuable contribution to the understanding of the present and future dynamics of African wetlands. However, since climate change is a consideration here, more of the current literature for Pan African needs to be included. The work presented supports the conclusions and claims, but there may be additional important work required to justify the data interpretation and conclusions (significance of trends). Notwithstanding, enough detail is provided on the methods in order to enable reproduction of the work.

1.Comment: Pan African climate change work needs to be included

Response: Thank you to the reviewer for providing literature related to climate change in Africa, which has helped address the gaps in our study regarding climatic aspects. We have taken the reviewer's advice to "incorporate more literature related to pan-African climate change." Consequently, we have added relevant content to the main text and the supplementary material in the discussion section, including the literature provided by the reviewer. The revised details can be found on [Page 11, lines 04-06]:

In the 21st century, the annual average temperature in some regions of Africa may rise at about 1.5 times the rate of global temperature increase^{2,8}, while the frequency and intensity of precipitation remain more uncertain^{3,9,10}.

Supplementary Discussion 2—Climate Change in Africa:

Climate change referred to the long-term transformation or alteration of climate at a specific location, region, or globally, resulting from either natural variations or human activities. Africa is the second most populous continent in the world, significantly impacted by climate change while having low adaptive capacity, making it one of

regions the most vulnerable to climate change.....

2.Comment: Page 4, line 3: Yes, but for some areas their small-scale losses may have huge ecological impacts. These impacts may be more visible if the study area is made smaller, e.g., only looking at the southern African region as opposed to the whole Africa.

Response: We fully agree with this point by the reviewer's perspective that the loss of wetlands in specific areas remains stark and cannot be overlooked. It's important to recognize that while there is an overall increase in Africa's wetlands, it does not mask the severity of local losses. In response to this issue highlighted by the reviewer, we have made additions to the main text to address this concern. The revised details can be found on [Page 16, lines 10-16]:

It is important to note that although Africa's wetlands have not experienced significant overall loss, there has been substantial reduction in certain regions (e.g., the Congo Basin, Southern Africa) and in coastal wetlands. The quality of Africa's wetlands still needs to be assessed, as stable or expanding wetland areas do not necessarily indicate a favorable condition. The degradation of the ecological functions of existing wetlands continues to pose serious threats to local communities, for example, the shrinking of Lake Chad's surface area.

3.Comment: Page 6, line 12: The decrease seems small over southern Africa, but is the ~2% loss from a Pan African perspective or from a southern African one?

Response: The reviewer seems to be puzzled by the approximately 2% loss ratio shown in Supplementary Figure 5, which depicts the wetland loss from 1984 to 2021 in southern African countries (such as South Africa, Botswana, Malawi, etc.).

Here, we will detail the calculation steps for this figure. We first gathered data on the changes in wetland areas for African countries between 1984 and 2021. If a country's wetland area decreased during this period, it was marked as a wetland loss country. The decrease in wetland area (loss area) for these countries was then summed to obtain the total wetland loss area for all of Africa. The loss ratio for each country was calculated by dividing its specific decrease in wetland area by the total loss area for the

continent. The same method was applied to calculate increases for countries where wetlands grew.

It is important to note that the magnitude of this ratio is directly related to the total wetland area in each country. Those countries like South Africa and Botswana have much smaller wetland areas compared to countries like the Congo or South Sudan. Therefore, regardless of the intensity of wetland area changes in these countries with smaller wetlands, their loss or gain ratios will not appear very high.

In summary, this approximately 2% ratio is viewed from a Pan-African perspective.

4.Comment: A number of figures in the main text and also the supplementary material indicates linear trends, but without showing what the statistical significance levels of these trends are, the trend results are irrelevant.

Response: Thank you very much to the reviewer for highlighting the areas of imprecision in our study. Following the reviewer's suggestion, we have revised and supplemented the figures in the main text and supplementary materials. The revised details can be found on [Page 07, lines 09-14]:

Fig. 2 | Changes and trends in the area of inland and coastal wetlands in Africa during the historical period. The black dotted line is the inland wetland change trend, and the red dotted line is the coastal wetland change trend. **The trend line is obtained**

by one-dimensional linear fitting in MATLAB, and P-value and Z was calculated. The horizontal coordinates were the middle years of 10 classification periods.

Supplementary Figure. 6 | Time Series and trends of area changes in African wetlands from 1984 to 2021. The linear trend was calculated since 2000. P-value and Z value were obtained by Mann-Kendall method.

Supplementary Figure. 8 | Comparison of HII between coastal wetlands area and inland wetlands area in Africa after 2000. HII was averaged over the same time span as the classification. The horizontal coordinates are the middle years of the seven periods after 2000, and the vertical coordinates are the HII values. **P-value was obtained by the Mann-Kendall method.**

References

- 1 Darrah, S. E. et al. Improvements to the Wetland Extent Trends (WET) index as a tool for monitoring natural and human-made wetlands. *Ecol. Indic.* 99, 294-298, doi:10.1016/j.ecolind.2018.12.032 (2019).
- 2 Almazroui, M. et al. Projected change in temperature and precipitation over Africa from CMIP6. *Earth Syst. Environ.* 4, 455-475 (2020).
- 3 Bobde, V., Akinsanola, A., Folorunsho, A., Adebisi, A. & Adeyeri, O. Projected regional changes in mean and extreme precipitation over Africa in CMIP6 models. *Environ. Res. Lett.* (2024).
- 4 Giannini, A., Biasutti, M., Held, I. M. & Sobel, A. H. A global perspective on African climate. *Clim. Change* 90, 359-383 (2008).
- 5 Zhang, Z. et al. Emerging role of wetland methane emissions in driving 21st century climate change. *Proceedings of the National Academy of Sciences* 114, 9647-9652 (2017).
- 6 Comyn-Platt, E. et al. Carbon budgets for 1.5 and 2 C targets lowered by natural wetland and permafrost feedbacks. *Nat. Geosci.* 11, 568-573 (2018).
- 7 Kleinen, T., Gromov, S., Steil, B. & Brovkin, V. Erratum: Atmospheric methane underestimated in future climate projections (2021 *Environ. Res. Lett.* 16 094006). *Environ. Res. Lett.* 16 (2021).
- 8 Engelbrecht, F. et al. Projections of rapidly rising surface temperatures over Africa under low mitigation. *Environ. Res. Lett.* 10, 085004 (2015).
- 9 Dosio, A. et al. Projected future daily characteristics of African precipitation based on global (CMIP5, CMIP6) and regional (CORDEX, CORDEX-CORE) climate models. *Clim. Dyn.* 57, 3135-3158 (2021).
- 10 Dosio, A., Lennard, C. & Spinoni, J. Projections of indices of daily temperature and precipitation based on bias-adjusted CORDEX-Africa regional climate model simulations. *Clim. Change* 170, 13 (2022).

Reviewer #4

General Comments: This is an interesting and ambitious paper that seeks to map trends in wetland extent over the period 1984-2021, and extrapolate forward to assess how climate change will impact wetlands in the future. The use of Landsat data over such a large region is impressive and the results are of general interest to a wider audience, making it potentially publishable in Nature Communications. However, I have a number of reservations about aspects of the work which in my opinion need to be addressed for the work to be published here.

Major comments

1. Evaluation of interannual variability against independent datasets

Figure 1b shows the change in African wetlands from 1984 to 2021. I am somewhat familiar with the Sudd wetland and was surprised to see areas of loss and gain in close proximity to each other. Hydrologically this seems rather curious, and in contrast to a very impressive study focused on this area by Hardy et al (2023). They document interannual variability in inundation over this period also using Landsat data. They demonstrate a remarkable expansion of the Sudd, particularly since 2019, and evaluate it against independent datasets. The Sudd is one of the largest wetlands in Africa, and the apparent misdiagnosis of the long-term trend here (with areas of loss and gain) raises serious concerns for me about the quality of the mapping. I would be reassured if the authors looked carefully at that study and evaluated their own results against that benchmark.

In a similar vein, I would like to see a comparison of the interannual variability presented in this study with the GIEMS dataset of wetlands (Bernard et al, 2024) across Africa. That dataset is at much coarser spatial resolution, bringing with it its own issues (e.g. missing small wetlands), but its interannual variability at the scale of such major wetlands should be credible. As it covers the period 1992-2020, it offers an independent dataset to evaluate the current methodology.

Response: We are most grateful to the editor for inviting a new expert to reassess our manuscript. Undoubtedly, the reviewer possesses deep familiarity and professional expertise in the field of African wetlands. The novel perspectives on wetlands they have introduced—such as methane emissions and GRACE-derived terrestrial water storage—have further enriched this study, and we express our sincere appreciation.

Reviewer Comment 1 raises two principal issues, which we address in turn.

1. The reviewer questions the apparent loss of Sudd wetland extent in our analysis and notes that the areas of loss and gain are in close proximity—a result that appears to conflict with Hardy et al. (2023).

2. The reviewer suggests that our study be compared with the GIEMS dataset to analyze interannual variations in African wetland area over the period 1992 - 2020.

Response to Issue 1

We have carefully studied Hardy et al. (2023) and the TropWet methodology they employed. We fully endorse both the conclusions and the methodological rigor of their work. We believe that the discrepancy between their findings and ours arises primarily from two factors:

1. Definition of wetland. TropWet extracts four principal wetland classes within the Sudd inundation zone: (1) inundated vegetation, (2) saturated soils, (4) open water, and (7) aquatic vegetation. The “inundated vegetation” category corresponds to vegetation flooded during the October - December inundation period, and thus encompasses areas that many studies would classify as grassland. In contrast, our study adopts a classification oriented toward stable wetland ecosystems or landscape patterns. We do not include grassland vegetation subject only to short - term flooding within our “herbaceous marsh” category. Consequently, although our extracted wetland extent is closely linked to hydrological dynamics, the development of a self-sustaining wetland ecosystem exhibits a temporal lag and does not coincide directly with inundation events. Moreover, our final temporal reference—2020, covering the 2019 - 2021 interval—coincides with the onset of Sudd flood expansion; we thus hypothesize that only multi-year inundation would yield a detectable increase in herbaceous marsh

extent under our definition.

2. Classification methodology. TropWet maximizes the influence of rainy-season hydrology (flooding and precipitation) on floodplain wetlands and adjacent grasslands, yielding strong correlations between their wetland extraction and GRACE - derived terrestrial water storage as well as Lake Victoria water level. By contrast, our approach seeks to discriminate wetlands from other upland cover types (grassland, forest). To achieve this, we incorporated a “flow distance” parameter as a simulated hydrological feature, thereby excluding most inundated grasslands (see red-boxed areas in Figure 2). Detailed procedures are available in our previous publication on African wetland classification.

Figure 1. Sude Wetland in 2020 (this study). Landsat imagery spanning 2019–2021; green areas denote inland marsh and yellow areas denote moist soil.

Figure 2. Inundated extent of Sudd Wetland based on TropWet data, October–December 2020.

In addition, to ensure classification reliability, we employed multi-year Landsat composites—using seven-year spans for the 1980s and three-year spans after 2000—rather than relying on just a few months of imagery from a single year as in the TropWet approach. While this strategy inevitably dampens our ability to capture intra-annual seasonal variations (for example, during the October–December (OND) short rains), and thus prevents us from precisely mapping wetland expansion within that specific season as TropWet does, our aim is to analyze long-term changes in the entire African wetland ecosystem. This trade-off was therefore an intentional design choice. In our results, the Sudd wetland shows a net loss between 1984 and 2021, which we attribute to the degradation and shrinkage of smaller riverine wetlands around its margins. Although extensive short-term flooding occurred after 2019, the newly inundated areas

lie mostly in the Sudd's core. The temporary submergence of surrounding grasslands does not meet our criteria for wetland expansion.

In summary, we agree with Hardy et al. (2023) that the inundated extent of the Sudd wetland expanded substantially during the short rainy seasons after 2019. However, based on our methodological framework and research focus, the extent of herbaceous marsh—defined as stable wetland vegetation—has remained relatively unchanged.

References:

Li A, Song K, Chen S, et al. Mapping African wetlands for 2020 using multiple spectral, geo-ecological features and Google Earth Engine[J]. ISPRS Journal of Photogrammetry and Remote Sensing, 2022, 193: 252-268.

Response to Issue 2

We downloaded and processed the GIEMS-MC dataset and also gathered the WAD2M dataset developed by Zhang et al. (2021) and the GWL-FCS30 wetland product released by Zhang et al. (2024) for comparative analysis.

Figure 3. Scatter charts comparing the current results against the African wetland flood extent from GIEMS-MC.

Figure 4. Scatter charts comparing the current results against the African wetland flood extent from GWL_FCS30.

Figure 5. Scatter charts comparing the current results against the African wetland flood extent from WAD2M.

The GIEMS-MC dataset covers the period from 1992 to 2020 and thus partially validates our pre-2000 findings. As illustrated, the interannual trajectory of African wetland area in GIEMS-MC closely mirrors our results: an initial decline followed by a subsequent increase. The greatest discrepancies occur in the 1980s, during which GIEMS-MC indicates only a slight decrease, whereas our analysis shows a more pronounced reduction. We attribute this divergence to the limited availability of Landsat

and other remote-sensing data in the 1980s; accordingly, both our study and prior work retain a degree of uncertainty regarding wetland extent during that decade, rendering the observed difference justifiable. In contrast, post-2000 comparisons yield more robust conclusions. Both GIEMS-MC and the 30 m-resolution GWL-FCS30 dataset clearly demonstrate an upward trend in African wetland area on interannual time scales. GWL-FCS30—a newly released global, 30 m time-series wetland dynamics product covering 2000–2022—exhibits interannual variations in African wetlands that align closely with our findings. By comparison, the WAD2M dataset does not display a consistent upward or downward trajectory and diverges significantly from the other two products.

References:

Zhang Z, Fluet-Chouinard E, Jensen K, et al. Development of the global dataset of wetland area and dynamics for methane modeling (WAD2M) [J]. Earth System Science Data, 2021, 13(5): 2001-2023.

Zhang X, Liu L, Zhao T, et al. Global annual wetland dataset at 30 m with a fine classification system from 2000 to 2022[J]. Scientific Data, 2024, 11(1): 310.

Conclusion

Finally, we contend that the differences between our study and the reviewer's referenced work arise primarily from divergent research foci and emphases. The wetland extent extracted in our study is neither more temporally responsive than the TropWet method nor more sensitive to flood inundation. However, with respect to more precisely delineating the boundary between wetlands and uplands and analyzing the gradual evolution of African wetland ecosystems, we are confident in the robustness of our results.

Interestingly, we observed that the reviewer's cited paper reports the global annual methane growth rate, whose interannual variability aligns more closely with the overall change in African wetland area documented in our study (and not solely with the Sudd wetland). Although our research did not explore the relationship between wetland area fluctuations and methane emissions directly, our findings may prompt researchers in

related fields to pay greater attention to changes in African wetlands.

[Figure Redacted]

Figure 2 in Hardy et al. (2023) illustrates: (A) the NOAA global annual methane growth rate (ppb); and (B) the OND (October–December) flood inundation extent computed by the TropWet model, expressed as a percentage anomaly relative to the 1984–2022 mean extent.

Figure 6. Comparison of the trend in African wetland area with the global annual methane growth rate.

2. Climate change, climate variability and the statistics of the rainfall-wetland relationship

The authors appear to have a view of African climate variability and change that emphasises the latter over the former. For example (P7 L1), they describe the time series of inland wetlands as “a strange trajectory”. In the context of strong interannual variability in climate (rainfall), particularly at the scale of large river catchments (e.g. Niger, Congo, Nile), I don't see this variability as strange at all. I would expect strong interannual variability in wetland extent dominating over any continental-scale rainfall trends. Instead (P1 L23) they state that “change of wetland area [is] closely related to climate change.”

The authors state (P10 L7-8) that “from 1984-2006... precipitation decreased significantly and continuously”, based on 4 time points. What accumulation period do these 4 time points actually represent? To justify their statement about “significant and continuous”, they should show interannual rainfall variability and test the rainfall trend over the period 1984-2006. The authors should also note the high degree of uncertainty in African precipitation trends across datasets (Maidment et al, 2015). The precipitation in the dataset that they use (ERA5) is not considered particularly reliable. In this context, a much better choice would be the ERA5-Land dataset (<https://www.ecmwf.int/en/forecasts/dataset/ecmwf-reanalysis-v5-land>), where they correct biases in the raw ERA5 precipitation.

Related to this, the authors state (P10 L5-9) that “the change in precipitation explains the violent fluctuation of wetland time series”, based on Figure 3b. The authors need to quantify how much of the variability the precipitation time series actually explains with a linear regression (and associated P-value). Given the lack of time points in their dataset, I would strongly recommend that they also perform a series of correlation analyses at sub-continental scale. Interannual rainfall variability is not spatially coherent at the continental scale, and consistent relationships between smaller-scale droughts and pluvials and wetland extent over effectively independent regions would greatly strengthen their arguments. Of course, it is reasonable to expect strong interannual rainfall variability to drive wetland extent, but Figure 3b does a poor job of

evidencing that relationship.

Response: We sincerely appreciate the reviewers for pointing out the shortcomings in our manuscript. We have humbly accepted the reviewers' suggestions and have provided explanations and corresponding revisions. In this set of comments, the reviewers primarily raised three key concerns:

1. The paper overlooks the high uncertainty in precipitation trends across different datasets. The reviewers requested evidence supporting the statement that "precipitation showed a persistent and significant decrease during 1984 – 2006..." and suggested validation using the ERA5-Land dataset.

2. The paper needs to quantitatively examine the relationship between precipitation and wetland fluctuations by employing linear regression (including p-values) to measure the explanatory power of the precipitation series. Additionally, the reviewers recommended supplementing the analysis with subcontinental-scale correlation assessments.

3. The paper neglects interannual climate variability and overattributes wetland changes to climate change.

Response to Issue 1

We sincerely thank the reviewers for providing the references. We have carefully reviewed the suggested literature and reintroduced multi-source precipitation data for further analysis. Regarding the reliability of the ECMWF_ERA5 dataset used in our study, the reviewers recommended validation using the ERA5-Land dataset. Accordingly, we processed the ECMWF_ERA5_LAND dataset and compared the results with those derived from ECMWF_ERA5. As shown in the figure, the two datasets exhibit no significant numerical differences. ECMWF_ERA5 employs the **total_precipitation** variable, while ERA5_LAND uses **total_precipitation_sum**.

Figure 7. Comparison of results between ECMWF_ERA5_MONTHLY and ECMWF_ERA5_LAND_MONTHLY. Monthly accumulated precipitation (**m**) in the major wetland distribution areas of Africa.

Regarding the comment that "precipitation trends exhibit high uncertainty across different datasets," we fully acknowledge the reviewers' concern and have expanded our analysis by incorporating 44 additional precipitation datasets, including:

- (1) ERA5-Land
- (2) GPCP (Global Precipitation Climatology Project)
- (3) TerraClimate
- (4) FLDAS

This multi-dataset approach ensures a more robust evaluation of precipitation trends and helps mitigate uncertainties associated with relying on a single data source.

Figure 8. Comparison of time series of four precipitation datasets. Monthly accumulated precipitation (**mm**) in the major wetland distribution areas of Africa.

The results indicate that the precipitation trends in the major wetland distribution areas of Africa did not exhibit a significant decline, except for ERA5_LAND (MK test: $P < 0.05$, $Z = -5.76$). This was an oversight in our study, and we have supplemented the main text accordingly. The revised details can be found on [Page 10, lines 05-15]:

Based on FLDAS-derived precipitation, the interannual variability of rainfall appears to account for the pronounced fluctuations observed in Africa's wetland extent time series. From 1984 to 2006, a sustained decline in precipitation coincided with a marked contraction of wetland area, whereas the subsequent rebound in rainfall was accompanied by wetland recovery. However, substantial uncertainties exist in Africa's interannual precipitation estimates across different datasets, and these uncertainties are spatially heterogeneous at the continental scale. We believe that although the amount of precipitation can directly affect the area of wetlands, evapotranspiration, runoff, and groundwater supply will also have an impact on local wetlands; therefore, the impact of precipitation on wetlands requires specific analysis of particular areas (more details in the Supplementary Information).

Reference:

Maidment, R. I., Allan, R. P., & Black, E. (2015). Recent observed and simulated changes in precipitation over Africa. *Geophysical Research Letters*, 42(19), 8155-8164. <https://doi.org/10.1002/2015gl065765>

Data availability

ERA5-Land is available from https://developers.google.com/earth-engine/datasets/catalog/ECMWF_ERA5_LAND_MONTHLY_AGGR.

ECMWF_ERA5_MONTHLY is available from https://developers.google.com/earth-engine/datasets/catalog/ECMWF_ERA5_MONTHLY.

GPCP is available from <https://www.ncei.noaa.gov/products/global-precipitation-climatology-project>.

TerraClimate is available from https://developers.google.com/earth-engine/datasets/catalog/IDAHO_EPSCOR_TERRACLIMATE.

FLDAS is available from <https://developers.google.com/earth->

engine/datasets/catalog/NASA_FLDAS_NOAH01_C_GL_M_V001.

Response to Issue 2

As the reviewers pointed out, we had underestimated the spatial complexity of interannual precipitation variability. Precipitation data over Africa still carry uncertainties, and the changes in African precipitation from 1980 to 2020 do not exhibit continental-scale spatial homogeneity. In response, we have supplemented our analysis by incorporating a region-specific assessment of the relationship between precipitation and wetlands at the subcontinental scale.

We primarily supplemented data from seven representative African wetlands (Fig. 9), examining the response relationship between wetland area and interannual precipitation variability:

- (1) Congo Basin
- (2) Inner Niger Delta
- (3) Lake Chad region
- (4) Sudd Wetland and surrounding areas
- (5) Zambezi River Basin
- (6) Lake Bangweulu
- (7) Okavango Delta

Figure 9. Classification results of representative African wetlands (2020)

The significance of interannual trends in wetland area (with missing data supplemented by linear interpolation) and precipitation in these regions from 1980 to 2020 was examined (Table 1).

Table 1. Trend analysis of wetland area changes

Region	S-statistic	Z-score	p-value	Sen's Slope	Significance
1	-542	-6.076479	1.23E-09	-663.4078	Significant
2	688	7.716341	1.2E-14	107.6894	Significant
3	566	6.346045	2.21E-10	167.5459	Significant
4	-584	-6.54822	5.82E-11	-270.0125	Significant
5	-80	-0.887323	0.374905	-3.077	Not significant
6	336	3.762699	0.000168	19.80628	Significant
7	-164	-1.830806	0.06713	-16.11553	Not significant

Table 2. Trend analysis of precipitation changes

Region	S-statistic	Z-score	p-value	Sen's Slope	Significance
1	-488	-5.469953865	4.5E-08	-0.613137	Significant
2	300	3.358349499	0.000784	0.229096	Significant
3	256	2.864144221	0.004181	0.245298	Significant
4	-248	-2.774288716	0.005532	-0.240429	Significant
5	202	2.257619563	0.023969	0.377895	Significant
6	130	1.448920018	0.14736	0.167962	Not significant
7	160	1.785878161	0.074119	0.259604	Not significant

It can be observed that, at the subcontinental scale, several representative wetland regions in Africa exhibit a predominantly significant monotonic trend in precipitation. Notably, the Congo Basin in Central Africa shows a significant downward trend, while the Inland Niger Delta in the Sahel region demonstrates a significant upward trend. Correspondingly, wetland areas in these regions—such as the forested wetlands of the Congo Basin and the herbaceous marshes of the Niger River—also exhibit significant decreasing and increasing trends, respectively, in response.

To further quantitatively assess the correlation between wetland area and

precipitation (monthly accumulated precipitation, in millimeters), the Spearman correlation coefficients and associated p-values were calculated. The results show that, among the seven subcontinental-scale wetlands selected for analysis, five exhibit statistically significant correlations between wetland area and precipitation, with moderate correlation strength ($0.3 \leq |\rho| < 0.7$). The wetlands of the Okavango Delta and Lake Bangweulu regions do not show significant correlations with precipitation variability. Moreover, none of the seven regions exhibit a strong correlation ($|\rho| \geq 0.7$) between wetland area change and precipitation.

Table 3. Spearman's rank correlation tests

Region	Wetland Area (km ²)	Precipitation (mm)	Spearman's ρ	p-value	Significance
1	296311.3479	145.1321746	0.680139373	2.15E-06	Significant
2	7708.180048	36.42429107	0.555052265	0.000211	Significant
3	15492.05352	38.61809293	0.425958188	0.005866	Significant
4	86741.74304	79.58088573	0.342160279	0.02909	Significant
5	5117.687515	75.65805395	0.330313589	0.035449	Significant
6	12513.2459	99.07811464	0.294076655	0.062359	Not significant
7	5318.688789	42.87566381	0.113240418	0.4795	Not significant

*Wetland area and precipitation values represent means across nine temporal periods

Response to Issue 3

We understand the reviewers' concerns regarding whether the current results are sufficient to attribute wetland changes to climate change, particularly in relation to precipitation.

Climate change refers to long-term shifts or alterations in climate at a specific location, region, or globally, resulting from either natural variability or human activities. Africa, the second most populous continent, is one of the regions most vulnerable to climate change due to its high exposure and low adaptive capacity (Sutton et al., 2011; Barros et al., 2014). According to the World Meteorological Organization's *State of the Climate in Africa 2023* report, the rate of warming in Africa slightly exceeds the global

average increase (<https://wmo.int/publication-series/state-of-climate-africa-2023>).

Although precipitation across Africa does not show a statistically significant overall increase or decrease, clear trends are evident in certain regions. As noted earlier in our subcontinental-scale correlation analysis between wetland area and precipitation across seven wetlands, precipitation significantly increased in three regions and significantly decreased in two. Furthermore, based on the Spearman correlation coefficients, changes in wetland areas—such as those in the Congo Basin (declining area), the Inland Niger Delta, and Lake Chad—are significantly positively correlated with precipitation.

In this study, we acknowledge that we may have overemphasized the analysis of climate factors (e.g., temperature, precipitation) at the continental scale in relation to wetland extent changes, while overlooking the complexity and regional variability of climate change across different parts of Africa. We concur with several of the reviewers' points, yet we also maintain that increases in precipitation can indeed contribute to the expansion of wetland areas. Accordingly, we have revised the manuscript to avoid broadly attributing wetland changes to overall precipitation trends and have instead focused on subcontinental-scale regional analyses. We sincerely appreciate the reviewers' professional and valuable suggestions.

Reference:

Sutton MA et al (2011) Summary for policy makers. Eur Nitrogen Assess.

<https://doi.org/10.1017/cbo9780511976988.002>

Barros VR et al (2014) Climate change 2014 impacts, adaptation, and vulnerability

Part B: regional aspects: working group II contribution to the fifth assessment report of the intergovernmental panel on climate change.

<https://doi.org/10.1017/CBO9781107415386>

Supplementary Fig. Comparative analysis of wetland area and precipitation trends in selected African wetlands

(1) Congo Basin

(2) Inner Niger Delta

(3) Lake Chad region

(4) Sudd Wetland and surrounding areas

(5) Zambezi River Basin

(6) Lake Bangweulu

(7) Okavango Delta

3. Cloud contamination

Apart from the brief mention in SI (P4) about the omission of the 1991-1995 period, there is no discussion of the problem of estimating wetland extent using Landsat in cloud-dominated regions (most notably the Congo Basin). At the very least, I would expect the paucity of cloud-free images in such areas would enhance the uncertainty in the resulting wetland mapping there. Is that not the case? I would like to see some evidence of that.

Response: Due to cloud cover, the reduction in usable Landsat imagery poses a

significant challenge for wetland classification, particularly in equatorial regions of Africa. This issue was indeed encountered in our study. Below, we describe our solutions in detail:

The first strategy involved extending the temporal span of imagery. Initially, we attempted to extract wetland areas in the Congo Basin using Landsat imagery composited over a single year. However, we found that this approach was insufficient for achieving complete spatial coverage in the region. This limitation was especially pronounced in pre-1990 Landsat imagery, where post-cloud-masking procedures resulted in severe data loss. To address this, we extended the temporal window (see Table 1). For the 1980s, we utilized imagery from 1984 to 1990—covering a total of seven years. During this period, we generated composite feature images by calculating spectral statistics (including maximum, minimum, and median values of key spectral bands, vegetation indices, and water indices), following the method of Murray et al. (2019). In subsequent periods, as the availability of Landsat imagery improved, the temporal window was progressively shortened to five years and eventually to three years.

Table 4. Number of Usable Landsat Images Across Ten Time Periods

Time Period	Temporal Span (Years)	Number of Images
1987	7	57951
1993	5	52944
1998	5	69605
2002	3	51431
2005	3	57024
2008	3	60176
2011	3	47695
2014	3	128781
2017	3	149039

Certainly, the inconsistency in the volume of imagery across different periods introduces uncertainty in classification accuracy. In 1987, the producer's and user's accuracies for wetland classification across Africa were only 77% and 91%, respectively. By 2020, these figures had improved to 85% and 93%. We consider cloud cover an uncontrollable factor in wetland classification. However, the measures implemented in this study have substantially increased the availability of usable Landsat imagery, and the resulting classification outputs can reasonably reflect the spatial distribution of wetlands in the 1980s to a certain extent.

A second method involved incorporating the “flow distance” feature—a hydrologically relevant wetland indicator derived from the D8 single-flow direction algorithm and hydrological process modeling. This approach follows the method proposed by Bwangoy et al. (2010) and significantly improves the accuracy and reliability of detecting forested wetlands (woody swamps) in the Congo Basin.

There is a strong spatial correlation between wetland distribution and river proximity. Wetlands are significantly distinguishable from upland ecosystems (e.g., forests, grasslands, and croplands) based on their nearness to rivers; the vertical distance from most wetland types to the nearest river is typically less than 10 meters. This pattern can be attributed to the mechanisms of wetland formation: areas within this buffer zone are generally flat and poorly drained, subject to periodic river inundation and groundwater recharge, creating stable wetland environments.

As a static terrain-hydrological feature invariant over time, we first applied a pit-filling process to eliminate minor topographic depressions in the surface raster, ensuring the smoothness of the terrain data. Following a comprehensive comparison of three flow-direction algorithms (D8, MFD, and DINF), and considering the hydrological characteristics required for arid-region wetlands, we ultimately selected the D8 single-flow-direction algorithm to construct the terrain flow-direction raster. This approach precisely determines the optimal flow path from each pixel to its downslope adjacent cell. Upon completing the flow-direction modeling, we generated progressively accumulating flow-accumulation rasters using the flow-accumulation algorithm,

accurately characterizing the spatial expansion patterns of surface runoff. Subsequently, a dual-threshold method was employed for stream network extraction, with particular focus on evaluating the differences in channel network density between two flow-accumulation thresholds: 1,000 and 20,000. The lower threshold (1,000) emphasizes the delineation of fine-scale tributary networks, while the higher threshold (20,000) targets the extraction of major channel structures. Through comparative analysis, this method effectively reveals the hierarchical characteristics of wetland drainage systems.

Cloud contamination is an unavoidable challenge in remote sensing – based wetland studies. We have applied the two methods described above to mitigate its impact as much as possible. Our results indicate that data from post-2000 are essentially unaffected by clouds, whereas for the pre-2000 period, the remaining uncertainty stems primarily from the limited availability of Landsat imagery.

Reference:

Murray, N.J., Phinn, S.R., DeWitt, M., Ferrari, R., Johnston, R., Lyons, M.B., Clinton, N., Thau, D., Fuller, R.A., 2019. The global distribution and trajectory of tidal flats. Nature, 565, 222–225.

Bwangoy J-R B, Hansen M C, Roy D P, et al. Wetland mapping in the Congo Basin using optical and radar remotely sensed data and derived topographical indices[J]. Remote Sensing of Environment, 2010, 114(1): 73-86.

4. Representation of fluvially-driven inundation

Clarification is needed for how TOPMODEL represents wetlands which are primarily driven by non-local rainfall. For example, interannual fluctuations in the Sudd and Niger Inland Delta are dominated by rain that fell hundreds of kilometres away and that spills out on the floodplain. I found it hard to infer how large a region contributes to the precipitation used in these areas from the description in SI section 8, which mentions over 40,000 watershed units. Would a map showing (at least the larger) watershed units help, or are they all too small to be represented on an African map? If the latter is the case, what does it mean to assume that TOPMODEL can simulate wetlands such as the

Sudd based only on local rainfall? I assume that assimilation of remotely-sensed soil moisture data in FLDAS implicitly captures fluvial-driven variability in such cases, but that source of information is not available in the climate change simulations, which do not depict the key hydrological processes.

Response: Thank you to the reviewer for your valuable comments regarding the use of TOPMODEL in our study to simulate wetlands primarily influenced by non-local precipitation sources, such as the Sudd Wetland and the Niger Inland Delta. Below is a detailed response to your concerns:

The methodological framework for wetland simulation in this study primarily follows the work of Yi Xi et al. (2021), published in Nature Climate Change, titled "Future impacts of climate change on inland Ramsar wetlands." The approach is outlined as follows:

[Figure Redacted]

Figure 10. Introduction to the TOPMODEL-based diagnostic model, adapted from Yi

Xi et al. (2021)

[Figure Redacted]

Figure 11. Calculation steps for water table depth (WTD), adapted from Yi Xi et al. (2021)

The classical TOPMODEL is a semi-distributed rainfall–runoff model based on the Topographic Wetness Index (TWI), originally proposed by Beven and Kirkby (1979). It was developed to simulate soil saturation and surface runoff processes under precipitation recharge. The model uses time series inputs of precipitation (along with other meteorological variables such as evapotranspiration and temperature) and estimates water table depth (WTD) and runoff by calculating infiltration and saturation-excess overland flow, thereby enabling dynamic simulation of wetland extent in response to hydrological changes.

In the TOPMODEL-based diagnostic model employed in Xi et al. (2022) and in this study, precipitation time series is not directly used as the driving input. Instead, multiple reanalysis soil moisture (SM) datasets are used (in place of precipitation), together with soil temperature and freeze–thaw status, to calculate the average water table depth (WTD) for each grid cell. Subsequently, based on the analytical relationship between the Compound Topographic Index (CTI), which reflects topographic control, and the WTD, the inundation threshold CTI^* is derived. This allows for the diagnosis of the proportion of inundated/saturated pixels (on a monthly basis) within each grid

cell, yielding the wetland fraction f_x , and thus enabling simulation of the spatiotemporal dynamics of wetlands.

In other words, this diagnostic algorithm "decouples" the explicit simulation of precipitation recharge in the original TOPMODEL by implicitly incorporating precipitation forcing into soil moisture reanalysis data, rather than directly treating precipitation as an independent driver. This approach is feasible for two main reasons:

(1) Land Data Assimilation Systems (LDAS) such as FLDAS indirectly account for the effects of runoff and river inflow on wetland water levels by assimilating remotely sensed soil moisture observations. As a result, the reanalyzed Water Table Depth (WTD) reflects variability driven by non-local precipitation.

(2) Although transient inundation driven by precipitation (e.g., flooded grasslands after heavy rainfall) falls under the broad definition of wetlands, its signal tends to be "averaged out" by short-term fluctuations in multi-decadal time series analysis. The model instead aims to capture long-term saturated zones that are more ecologically significant in shaping ecosystem patterns. Thus, using soil moisture reanalysis instead of direct precipitation inputs not only simplifies computation but also emphasizes persistent wetland dynamics.

Upon revisiting the manuscript, we realized that we had erroneously referred to the simulation method as "TOPMODEL," when in fact it should be described as a "TOPMODEL-based diagnostic model." This oversight stemmed from an insufficiently nuanced understanding of the TOPMODEL methodology, and we sincerely appreciate the reviewer for bringing this to our attention. We have corrected this in both the main text and supplementary materials.

Furthermore, we acknowledge that while the TOPMODEL-based diagnostic model efficiently utilizes the compound topographic index (CTI) and reanalyzed soil moisture (SM) to estimate long-term saturation and inundation fractions at the grid scale, it does omit explicit simulations of river routing and floodplain processes, nor does it directly incorporate precipitation time series as input. Additionally, the model cannot resolve microtopography and small-scale wetland dynamics, nor does it account for anthropogenic regulation (e.g., dams, irrigation). Consequently, it exhibits notable

limitations in simulating floodplains primarily driven by riverine inputs or extreme precipitation (e.g., the Sudd Wetland). Future improvements should involve coupling with river routing models, employing higher-resolution DEMs, and integrating direct precipitation forcing.

Reference

Stocker B D, Spahni R, Joos F. *DYPTOP: a cost-efficient TOPMODEL implementation to simulate sub-grid spatio-temporal dynamics of global wetlands and peatlands*[J]. *Geoscientific Model Development*, 2014, 7(6): 3089-3110.

Xi Y, Peng S, Ciais P, et al. *Future impacts of climate change on inland Ramsar wetlands*[J]. *Nature Climate Change*, 2021, 11(1): 45-51.

Minor points

Minor Point 1

P3 L2 and throughout: The authors refer to “10 periods from 1984 to 2021” though only 9 points are shown in Figures 2 and 3. In SI (P4) they note that the period 1991-5 suffered from lack of data and presumably this explains its absence in those figures. For consistency in the main text, it would make sense to talk about analysis of 9 periods, not 10.

Response: We have carefully addressed the reviewer's comments and made the following revisions to the manuscript.

[Page 03, line 02]:

Based on these sampling points, we used the random forest classifier to classify and extract African wetlands in 9 periods from 1984 to 2021 on the Google Earth Engine (GEE) platform.

[Page 05, line 04]:

Fig. 1 | Area distribution and changes of African wetlands during the historical period. a, The distribution of wetland areas in Africa from 1984 to 2021 (The average wetland area in 9 periods).

[Page 07, line 14]:

The horizontal coordinates were the middle years of 9 classification periods.

[Page 18, line 25]:

A total of 9 wetland maps were generated from the classification of African wetlands in historical periods (more details in the Supplementary Information).

[Page 19, line 07]:

The data regarding temperature, precipitation, PDSI, SM, and HII were uniformly resampled to a resolution of 30 m, whereupon the mean values of the main distribution areas of wetlands in 9 periods were calculated.

.....

They are not listed here individually.

Minor Point 2

The authors overstate the case with respect to the historical trend in inland wetland (P6 L19) “completely opposite trends... in coastal and inland wetlands” . Whilst the downward trend in coastal wetland is statistically robust, the inland trend ($P=0.175$) cannot be distinguished from zero at even the 10% level.

Response: We have carefully addressed the reviewer's comments and made the following revisions to the manuscript. The revised details can be found on [Page 06, line 19]:

The 38-year time span allowed us to study African wetlands in terms of time series and trends. Accordingly, we discovered different trends when comparing changes in the areas of coastal wetlands and inland wetlands in Africa from 1984 to 2021 (Fig. 2; Supplementary Fig. 6).

Minor Point 3

P8 L10 “has been conversion to...”

Response: We sincerely thank the reviewer for the detailed and careful review. We have corrected this oversight accordingly. The revised details can be found on [Page 08, line 10]:

The primary direct driver for the loss of coastal wetlands **has been converted to** other land uses (aquaculture, agriculture, coastal developments, etc.), which is the general consensus of the international community^{29,33,36}.

Minor Point 4

I may have missed it, but what soil moisture levels (in FLDAS and the CMIP6 models) were actually used? On a related note, the y-axis range in SI Fig. 20 (~650 m³/m³) is clearly wrong.

Response: Thank you to the reviewer for identifying this issue. The supplementary figure SI Fig. 20 indeed reflects an oversight on our part, as we did not clearly explain the methodology.

The FLDAS soil moisture data are provided in four layers and expressed in volumetric units (volume fraction). In our analysis, we first converted these volumetric data into units of kg/m². We then summed the soil moisture (kg/m²) across the four depth intervals 0–10 cm, 10–40 cm, 40–100 cm, and 100–200 cm to obtain the total mass of water in the 0–200 cm soil profile at each location. This value represents the soil water storage, i.e., the total amount of water contained per unit area within the soil profile. The same processing method was applied to the CMIP6 model outputs.

```
var SM_1 = SoilMoi0_10cm.multiply(1000).multiply(0.1);  
var SM_2 = SoilMoi10_40cm.multiply(1000).multiply(0.3);  
var SM_3 = SoilMoi40_100cm.multiply(1000).multiply(0.6);  
var SM_4 = SoilMoi100_200cm.multiply(1000).multiply(1);  
var SM = SM_1.add(SM_2).add(SM_3).add(SM_4);
```

Figure 12. Calculation of Total Soil Moisture in GEE

The error in the original figure lies in the unit label, which was incorrectly written as m³/m³ instead of the correct unit, kg/m². We have revised both the figure and the corresponding textual descriptions accordingly. The revised details can be found in the Supplementary Information [Page 42, lines 02-04]:

Supplementary Figure. 20 | Comparison of CMIP6 soil moisture time series and FLDAS soil moisture change trend. The FLDAS soil moisture represents the sum of soil moisture (kg/m^2) across four layers: 0–10 cm, 10–40 cm, 40–100 cm, and 100–200 cm, reflecting the total amount of water contained in the soil per unit area. The CMIP6 soil moisture time series is derived by averaging the soil moisture data from 14 models.

Additionally, the slightly elevated values observed in the figure are due to the exclusion of arid regions—such as North Africa, the Sahara Desert, and southern Africa—where wetlands are sparse. The statistics were calculated based solely on the major wetland distribution regions in Africa (see Supplementary Figure 1), which explains the relatively higher total soil moisture values.

Table 5. FLDAS Layered Soil Moisture Data.

Name	Units	Description
SoilMoi00_10cm_tavg	Volume fraction	Soil moisture (0 - 10 cm underground)
SoilMoi10_40cm_tavg	Volume fraction	Soil moisture (10 - 40 cm underground)
SoilMoi100_200cm_tavg	Volume fraction	Soil moisture (100 - 200 cm underground)
SoilMoi40_100cm_tavg	Volume fraction	Soil moisture (40 - 100 cm underground)

FLDAS is available from https://developers.google.com/earth-engine/datasets/catalog/NASA_FLDAS_NOAH01_C_GL_M_V001#bands.

Minor Point 5

SI Figure 19 shows that TOPMODEL forced by FLDAS soil moisture does a good job

at representing interannual wetland variability. Can the authors clarify how the calibration procedure (to determine the value of the M parameter) has influenced this simulation? In other words, was there data held back from the calibration step to evaluate the model simulations?

Response: The reviewer raised concerns regarding the calibration process of the wetland simulation, specifically the method used to determine the parameter M. We offer the following clarification:

As illustrated in the figure below, the study “Future impacts of climate change on inland Ramsar wetlands” defines M as an adjustable parameter ranging from 1 to 15, used to describe the exponential decay of soil permeability with depth and to capture regional heterogeneity.

[Figure Redacted]

In our study, we treated the M value in each grid cell as a fixed attribute. We systematically tested all values from 1 to 15, calculating the root mean square error (RMSE) between the observed wetland area (mean across nine historical periods) and the simulated wetland area (mean over the same nine periods). The M value that

minimized RMSE was selected as the optimal parameter for that grid cell and was subsequently used for future wetland simulations.

Supplementary Figure. 19 | Comparison of simulated wetland area time series and classified wetland area time series. The left coordinate represented the classified inland wetland area, and the right ordinate represented the simulated inland wetland area based on FLDAS soil moisture data. The vertical coordinates were the middle year of 10 periods.

We understand the reviewer's concern that calibrating M in this manner could potentially lead to overfitting to the classified data. However, it is important to clarify that M was calibrated only once, based on the mean wetland extent across the nine historical periods, and was not fitted separately for each period. Therefore, the chosen M value remains constant across both historical and future simulations. This fixed parameter influences only the overall magnitude of simulated wetland area and does not affect its interannual variability or temporal dynamics.

Regarding the question of whether any data were withheld during the calibration step to independently evaluate model performance: only the simulation results using the optimal M values (post-calibration) were retained, as shown in Supplementary Figure 19. Simulations with other M values were not subjected to further analysis.

Reference in the figure:

[25] Beven K J, Kirkby M J. A physically based, variable contributing area model of basin hydrology/Un modèle à base physique de zone d'appel variable de l'hydrologie du bassin versant[J]. *Hydrological sciences journal*, 1979, 24(1): 43-69.

Minor Point 6

In Figure 5, what range of values does a “constant” wetland area correspond to in the bar charts? In the maps, it would be better to depict that “constant” orange over a range at equal distance either side of zero (e.g. -3 to +3).

Response: We sincerely thank the reviewer for pointing out the overly strict classification of "stable" wetland areas in Figure 5 and for suggesting the use of an orange (or yellow) color on the map to indicate regions with changes within a ± 3 km² range.

In the original manuscript, the "stable" wetlands shown in the bar chart of Figure 5 referred only to pixels with an exact change of zero in wetland area. Clearly, this zero-threshold definition excluded minor fluctuations and resulted in an overly conservative estimate of stability, yielding too few samples in the “stable” category.

We have adopted the reviewer’s suggestion by redefining "stable" wetland areas as those with a change in area within ± 3 km². These regions are now highlighted in yellow on the map. Based on this revised threshold, we recalculated the proportions of wetland area that fall into the categories of increase, decrease, and stable under the four scenarios: SSP1-2.6, SSP2-4.5, SSP3-7.0, and SSP5-8.5. Corresponding updates were made to the values and color scheme in the pie charts in Figure 5.

The revised figure is as follows:

Fig. 5 | Inland wetland area changes in Africa by 2100 under four SSP scenarios mapped per 0.2 ° grid cell. a-d are inland wetland changes under SSP126, SSP245, SSP370, and SSP585 scenarios, respectively. The pie chart shows the proportion of areas with inland wetland area changes in Africa.

The corresponding textual description in the main text, under the section “Inland wetland trends under future climate trajectories,” has also been revised accordingly and highlighted in the revised manuscript. The revised details can be found on [Page 12, lines 14-15]:

In the SSP126 scenario, which features more moderate climate change than the others, 17% of Africa’s regions will experience wetland area growth; in comparison, the SSP585 scenario predicts that 35% of the regions will experience wetland area expansion (Fig. 5).

Reviewer #4 (Remarks to the Author):

General Comments: The authors have provided a lot of material in response to my comments! Thank you for that considerable effort. Below I flag some areas where I am still unclear about their logic. On several points, the authors do not appear to have made any change to the manuscript. I would like to see updates in the manuscript consistent with evidence presented in their response. It may be that I missed some changes in the manuscript, but lines featuring yellow highlighting in the revised manuscript are remarkably few given the 35-page response.

General response: We appreciate the reviewers' positive recognition of our previous revision, which has strengthened our confidence in the current study.

Given that the last revision already included extensive additional material, the present revision focuses on addressing the specific parts of the manuscript (main text and Supplementary Information) that remained unmodified, while keeping our responses concise. Moreover, considering Nature Communications' emphasis on logical flow and brevity in the main text, we have placed most of the detailed revisions in the Supplementary Information file and hope the reviewers will understand this approach.

Major point 1

The authors have done a good job of comparing their mapping with the TropWet dataset of Hardy et al, and arguing why their analysis differs from that study. I also appreciate their efforts to provide additional evaluation, particularly with respect to the GIEMS dataset. However, there are several points I must query them on. In their response they state (p5) "As illustrated, the interannual trajectory of African wetland area in GIEMS-MC closely mirrors our results: an initial decline followed by a subsequent increase." I agree that "the subsequent increase" in GIEMS is very clear in their response Figure 3. However, it is quite misleading to describe the earlier part of the GIEMS time series as

“an initial decline”. I struggle to see how the comparison “partially validates our pre-2000 findings”. I believe the authors when they argue that the limited availability of Landsat and other data pre-2000 affects the quality of the mapping. I think the authors need to add a sentence in the main text. I also think that the comparison they show in the response Figure 3 should be noted in the text and included in SI, though given the strong interannual variability in the GIEMS dataset, I don’t really see the value of the polynomial fits they present. I am also somewhat surprised that the authors don’t provide a correlation metric between the two datasets as, by eye, the comparison seems to provide important independent evaluation of their findings.

Response: We agree with the reviewer’s suggestion to clarify in the main text the impact of limited pre-2000 data on mapping quality, and to explicitly include and present the GIEMS comparison provided in our response within the Supplementary Information. The revised details can be found in the manuscript on [Page 07, lines 06 - 11]:

It should be noted that due to missing remote sensing imagery and cloud contamination, the classification results of this study differ from existing datasets such as GIEMS, WAD2M, and GWL-FCS30. Consequently, current research conclusions—particularly for periods before 2000—remain subject to uncertainty. A detailed comparison between our results and these datasets, along with explanations of the differences, is provided in the Supplementary Information.

Supplementary Information on [Page 05, lines 01 - 17]:

We compared our results with GIEMS-MC, GWL-FCS30, and WAD2M (Supplementary Figs. 22 - 24). The comparisons for the post-2000 period yield conclusions that can be regarded as robust. Both GIEMS-MC and the 30-m GWL-FCS30 clearly indicate an upward trend in African wetland area. By contrast, WAD2M shows no clear upward or downward trend and differs substantially from the other two datasets. GWL-FCS30, released in 2024, is the most recent global 30-m time-series wetland dynamics dataset and covers 2000 - 2022; its record of interannual variability in African wetlands is broadly consistent with our findings. The GIEMS-MC dataset

spans 1992 – 2020 and has a moderate positive linear correlation with our time series (Pearson $r = 0.421$, $p = 0.013$). The largest discrepancies between GIEMS-MC and our results are concentrated before 2000: GIEMS-MC shows no significant trend change, whereas our results indicate a more pronounced decline.

Because Landsat and other source data are sparse prior to 2000, current assessments (including ours) of pre-2000 African wetland conditions remain uncertain, making these discrepancies understandable. Whether African wetland area before 2000 experienced a true declining trend or merely normal interannual fluctuations still requires further study for confirmation.

Supplementary Figure. 22 | Scatter charts comparing the current results against the African wetland flood extent from GIEMS-MC. Missing years in the study dataset were supplemented by linear interpolation. The trend line represents a linear fit of GIEMS-MC data from 2000 to 2020.

Supplementary Figure. 23 | Scatter charts comparing the current results against the African wetland flood extent from GWL_FCS30.

Supplementary Figure. 24 | Scatter charts comparing the current results against the African wetland flood extent from WAD2M.

References:

1. Zhang Z, Fluet-Chouinard E, Jensen K, et al. Development of the global dataset of wetland area and dynamics for methane modeling (WAD2M) [J]. *Earth System Science Data*, 2021, 13(5): 2001-2023.
2. Zhang X, Liu L, Zhao T, et al. Global annual wetland dataset at 30 m with a fine classification system from 2000 to 2022[J]. *Scientific Data*, 2024, 11(1): 310.

Major point 2

Firstly, I must apologise for having provided some misleading information in my original review. As I have subsequently learned, precipitation in ERA5-Land is not in fact bias-corrected with observations. In that context, the similarity between ERA5_Land and ERA5 in the response Figure 7 is not surprising. Their response Figure 8 is rather revealing of the general problem that I raised on the other hand, with ERA5-Land showing a quite distinct negative trend prior to 2000 compared to the other datasets. Having gone to some effort to consider 3 other datasets, I am puzzled why they have not updated the main text consistently. On p10 (L7-8) they state that “From 1984-2006 there is a sustained decline in precipitation”. This statement is apparently based on the FLDAS dataset yet it looks at odd with FLDAS in response figure 8. Are the points in Figure 3 based on single year values from the response figure 8? That would be curious given the large interannual variability evident in response figure 8. Also the inconsistency in the 1987 value of soil moisture (and PDSI) compared to precipitation makes me wonder if they have plotted ERA5 precipitation data in Figure 3. In any case, response figure 8 provides no support for the apparent “sustained decline” if one ignores (or perhaps even weights equally) ERA5-Land.

The authors have done a lot of welcome additional regional analysis (their “issue 2”) of precipitation and wetland area datasets in response tables 1-3 and associated response figures. I am curious though about some of the very low p-values reported given the small number of data points ($n=9$ for wetland area trends). I think the regional analysis provides important supporting evidence that wetland area is responding to changes in precipitation. However, in the revised manuscript the authors have not adequately responded to my comments about “climate change”. The vast majority of readers will interpret the statement in the abstract “the change of inland wetland area is closely related to climate change” as meaning long-term anthropogenic climate change, a message reinforced by the phrase in the title “on the rise during the 21st century”. There is still no reference in the text to the importance of regional-scale decadal variability in precipitation.

Response: We apologize for a clerical error in our previous revision: the precipitation data plotted in Figure 3 were derived from ERA5 rather than FLDAS. From the outset we have stated that the precipitation data analyzed and shown in the manuscript originate from ERA5. In the last revision we preserved the original figure and explicitly noted that the precipitation time series came from a single product, and we added that “substantial uncertainties exist in Africa’s interannual precipitation estimates across different datasets, and these uncertainties are spatially heterogeneous at the continental scale.” The reviewer correctly indicated that this was insufficient.

Accordingly, in the current revision we have replaced the ERA5 precipitation shown in Figure 3 with the ensemble mean of three independent gridded products (TerraClimate, FLDAS and GPCP). Figure 3 and all related statements have been updated. The revised details can be found in the manuscript on [Page 10, lines 04 – 16]:

Interannual precipitation in Africa exhibits substantial uncertainty across different data products⁴³, and the spatiotemporal patterns of precipitation from 1984 to 2021 are not consistent at the continental scale. Notably, decadal-scale variability plays an important role in explaining precipitation trends and the episodic (phase-like) changes in wetland extent: the episodic increases and decreases of wetland area in some regions can be partly attributed to internal climate oscillations on decadal or longer timescales rather than to sustained linear trends. Although some regions show moderate correlations between wetland area and precipitation, this relationship is highly spatially heterogeneous and temporally non-stationary (see Supplementary Information). Moreover, hydrological processes such as evapotranspiration, surface runoff, and groundwater recharge also modulate the effect of precipitation on local wetlands⁴⁴⁻⁴⁶; therefore, integrated, region-specific analyses are required.

[Page 22, lines 27 – 28]:

precipitation data are available from https://developers.google.com/earth-engine/datasets/catalog/IDAHO_EPSCOR_TERRACLIMATE, https://developers.google.com/earth-engine/datasets/catalog/NASA_FLDAS_NOAH01_C_GL_M_V001, and <https://www.ncei.noaa.gov/data/global-precipitation-climatology-project-gpcp->

monthly/access/.

Fig. 3 | Time series comparison of wetland area and climate drivers in Africa from 1984 to 2021. a-c are the monthly average air temperature at 2m height (K), monthly total precipitation (m), palmer drought severity index (PDSI), and 0-200 cm underground soil moisture (kg/m²) in the main distribution areas of African wetlands.

Supplementary Information on [Page 16 - 18]:

10. Analysis of the Correlation Between Multi-Source Precipitation Data and Wetland Extent at the Subcontinental Scale

Interannual variability of African precipitation exhibits a high degree of uncertainty across different datasets, and spatially the changes in African rainfall from 1980 to 2020 do not display continent-wide homogeneity. Therefore, we have supplemented our study with a Spearman correlation analysis between precipitation and wetland area at the subcontinental scale.

We employed four precipitation datasets: (1) ERA5-Land; (2) GPCP (Global Precipitation Climatology Project); (3) TerraClimate; and (4) FLDAS. Seven representative subcontinental wetlands in Africa were selected: (1) the Congo Basin; (2) the Inner Niger Delta; (3) the Lake Chad region; (4) the Sudd Wetland and surrounding areas; (5) the Zambezi River Basin; (6) Lake Bangweulu; and (7) the Okavango Delta. Their precise extents are illustrated in the Supplementary Figure. 21.

We assessed the significance of the interannual trends in both wetland area (with missing values linearly interpolated) and precipitation (using the mean of the four datasets) over the 1980–2020 period (Supplementary Table 6). At the subcontinental scale, most of these characteristic wetland regions in Africa displayed significant monotonic trends in precipitation. In particular, the Congo Basin in central Africa exhibited a significant decreasing trend, whereas the Inner Niger Delta in the Sahel region showed a significant increasing trend. Correspondingly, the wetland areas in these regions—such as the forest wetlands of the Congo Basin and the herbaceous marshes of the Niger River—demonstrated significant declines and increases, respectively.

To quantitatively evaluate the relationship between wetland area and monthly cumulative precipitation (mm), we computed Spearman's rank correlation coefficients (ρ) and associated p-values. Among the seven selected subcontinental wetlands, five exhibited statistically significant correlations of moderate strength ($0.3 \leq |\rho| < 0.7$) between area and precipitation. The Okavango Delta and Lake Bangweulu wetlands did not show significant correlations with precipitation changes. Moreover, none of the seven regions demonstrated a strong correlation ($|\rho| \geq 0.7$) between wetland area variation and precipitation.

ERA5-Land is available from <https://developers.google.com/earth->

[engine/datasets/catalog/ECMWF ERA5 LAND MONTHLY AGGR](https://developers.google.com/earth-engine/datasets/catalog/ECMWF ERA5 LAND MONTHLY AGGR). GPCP is available from <https://www.ncei.noaa.gov/products/global-precipitation-climatology-project>. TerraClimate is available from https://developers.google.com/earth-engine/datasets/catalog/IDAHO_EPSCOR TERRACLIMATE. FLDAS is available from https://developers.google.com/earth-engine/datasets/catalog/NASA_FLDAS_NOAH01_C_GL_M_V001.

Supplementary Figure. 21 | W Wetland extent analyzed at the subcontinental scale in Africa, with the background displaying the 2020 wetland classification results.

Supplementary Table 6 | Correlation Between Precipitation and Wetlands in Africa.

(1) Trend analysis of wetland area changes

Region	S-statistic	Z-score	p-value	Sen's Slope	Significance
1	-542	-6.076479	1.23E-09	-663.4078	Significant
2	688	7.716341	1.2E-14	107.6894	Significant
3	566	6.346045	2.21E-10	167.5459	Significant
4	-584	-6.54822	5.82E-11	-270.0125	Significant
5	-80	-0.887323	0.374905	-3.077	Not significant
6	336	3.762699	0.000168	19.80628	Significant
7	-164	-1.830806	0.06713	-16.11553	Not significant

(2) Trend analysis of precipitation changes

Region	S-statistic	Z-score	p-value	Sen's Slope	Significance
1	-488	-5.469953865	4.5E-08	-0.613137	Significant
2	300	3.358349499	0.000784	0.229096	Significant
3	256	2.864144221	0.004181	0.245298	Significant
4	-248	-2.774288716	0.005532	-0.240429	Significant
5	202	2.257619563	0.023969	0.377895	Significant
6	130	1.448920018	0.14736	0.167962	Not significant
7	160	1.785878161	0.074119	0.259604	Not significant

(3) Spearman's rank correlation tests

Region	Wetland Area (km ²)	Precipitation (mm)	Spearman's ρ	p-value	Significance
1	296311.3479	145.1321746	0.680139373	2.15E-06	Significant
2	7708.180048	36.42429107	0.555052265	0.000211	Significant
3	15492.05352	38.61809293	0.425958188	0.005866	Significant
4	86741.74304	79.58088573	0.342160279	0.02909	Significant
5	5117.687515	75.65805395	0.330313589	0.035449	Significant
6	12513.2459	99.07811464	0.294076655	0.062359	Not significant
7	5318.688789	42.87566381	0.113240418	0.4795	Not significant

*Wetland area and precipitation values represent means across nine temporal periods

Regarding climate change, the reviewer mainly questioned our analysis of the period from the 1980s to 2020, arguing that the observed variations in Africa's climate were primarily interannual fluctuations without a significant long-term trend, and therefore should not be described as climate change (emphasizing decadal variability in precipitation).

This may be a misunderstanding. In our study, the conclusions about African climate change are based on CMIP6 projections for the period 2015–2100. As we mentioned in our response to the first reviewer, the latest CMIP6 simulations provide different projections. Studies have shown (Almazroui et al., 2020; Bobde et al., 2024) that mean precipitation over Africa is projected to increase under several scenarios—for example, by +6.2% under SSP1-2.6, +6.8% under SSP2-4.5, and +9.5% under SSP5-8.5. Except for southern Africa and Madagascar, most regions of the continent are expected to experience a fivefold increase in extremely wet years, particularly in the Sahel, where mean annual precipitation is projected to rise. Furthermore, a paper published in *Nature Ecology & Evolution*, “Human population growth offsets climate-driven increase in woody vegetation in sub-Saharan Africa,” noted that rising atmospheric CO₂ and precipitation have led to a greening trend even in African drylands.

Under different SSP scenarios (SSP1-2.6, SSP2-4.5, SSP3-7.0, and SSP5-8.5), we simulated African inland wetlands using soil moisture data from 14 CMIP6 models. The results (Fig. 4) show that, regardless of the scenario, the majority of models indicate an increasing trend in African inland wetland area, which underlies the title “African inland wetland area on the rise during the 21st century.”

Of course, we also agree with the reviewer that observations during the historical period show no significant long-term increase or decrease in African precipitation—only normal interannual fluctuations, as we have clearly stated in the manuscript.

In summary, our conclusions regarding African climate change are well supported. Based on CMIP6 projections, Africa is expected to experience long-term climatic changes during the 21st century (2015–2100), characterized by increases in both temperature and precipitation. Our simulations of African inland wetlands using CMIP6 data and the TOPMODEL framework therefore reflect the impacts of long-term

anthropogenic climate change (under different SSP scenarios), which is projected to lead to an overall expansion of wetland area. Hence, the title and abstract of our paper are appropriate and unlikely to cause any misunderstanding.

References:

1. *Almazroui M, Saeed F, Saeed S, et al. Projected change in temperature and precipitation over Africa from CMIP6[J]. Earth Systems and Environment, 2020, 4: 455-475.*
2. *Bobde V, Akinsanola A A, Folorunsho A H, et al. Projected regional changes in mean and extreme precipitation over Africa in CMIP6 models[J]. Environmental Research Letters, 2024.*
3. *Brandt M, Rasmussen K, Peñuelas J, et al. Human population growth offsets climate-driven increase in woody vegetation in sub-Saharan Africa[J]. Nature ecology & evolution, 2017, 1(4): 0081.*

The revised details can be found in the Supplementary Information on [Page 03 – 04]:

2. Climate change in Africa

Climate change referred to the long-term transformation or alteration of climate at a specific location, region, or globally, resulting from either natural variations or human activities. Africa is the second most populous continent in the world, significantly impacted by climate change while having low adaptive capacity, making it one of regions the most vulnerable to climate change. Particularly, Africa's agricultural sector, which employs over 60% of the population, was highly sensitive to changes in temperature and precipitation. Climate change (e.g., the increased frequency of extreme weather events) was expected to have significant socioeconomic impacts on Africa. Future climate data from the latest Phase 6 of the Coupled Model Intercomparison Project (CMIP6) had become available for analysis²⁵. The CMIP6 comprised multiple climate models, incorporating simulation results from the world's leading climate research institutions. This multi-model ensemble approach allowed researchers to assess inter-model uncertainties and enhance the reliability of predictions through integrated analysis²⁶.

Some studies indicated that the CMIP6 predicted an increase in precipitation and a rise in temperature across most of Africa²⁷⁻³⁰. In contrast, Southern Africa and Madagascar were projected to become drier in the future³¹. According to the multi-

model median, the frequency of extremely wet years was expected to increase fivefold between 2050 and 2100 in most parts of Africa, excluding Southern Africa and Madagascar²⁹. Specifically, in the Sahel region, research found that the past two decades (1991-2010) were characterized by a slight wetting trend, with a reduction in the geographic extent affected by drought, while the annual average precipitation in the 21st century was projected to increase³².

In this study, we also used the CMIP6 data of temperature, precipitation, and soil moisture to evaluate future climate change in Africa. Our conclusions were largely consistent with previous studies. As shown in Supplementary Figure 14 and 15, temperature, precipitation, and soil moisture were projected to increase across most of Africa during the 21st century. The primary areas of temperature increase were concentrated in the regions north of the Sahara (such as Morocco and Algeria) and in Southern Africa (including South Africa, Botswana, and Angola). The changes in precipitation displayed a marked north-south contrast across Africa. The Congo Basin experienced a significant decrease in precipitation, while the areas in Southern Africa where precipitation decreased corresponded closely with regions of rising temperatures. Future changes in soil moisture were expected to manifest as increases in the Sahel and the Horn of Africa, with decreases in Central and Southern Africa.

Supplementary Figure. 14 | The spatial distribution of temperature and precipitation changes by 2100 under the SSP126 scenario.

Supplementary Figure. 15 | The spatial distribution of soil moisture changes by 2100 under the SSP126 scenario.

Supplementary Figure. 25 | Time series of projected temperature changes in Africa under different SSP scenarios

Supplementary Figure. 26 | Time series of projected precipitation changes in Africa under different SSP scenarios.

Supplementary Figure. 27 | Time series of projected soil moisture changes in Africa under different SSP scenarios.

Major point 3

The authors provide a very lengthy discussion of issues related to cloud contamination in their response. This concludes with a statement that “the data from post-2000 are essentially unaffected by clouds”. Did I miss the evidence for this, or is it simply an assertion? I suggest that the earlier comparison with GIEMS data (unaffected by cloud) does provide evidence, but they don’t mention it here. There is still no mention of cloud issues in the text as far as I can see.

Response: We thank the reviewer for raising concerns about cloud contamination. The reviewer acknowledges that cloud-related uncertainty affects wetland classification before 2000, but notes that evidence was not provided to show that data after 2000 are essentially free from cloud effects.

In our previous reply we explained that cloud impacts were mitigated by multi-year image compositing and by incorporating topographic and hydrological predictors. The Supplementary Information also presents the spatial distribution of good-quality observations within individual pixels of Landsat 4/5/7/8 images for ten periods from 1984 to 2021 across Africa (Supplementary Fig. 16).

Supplementary Figure. 16 | Spatial distribution of good-quality observations within individual pixels of Landsat 4/5/7/8 images in 10 periods from 1984 to 2021 in Africa.

The reviewer primarily requested a comparison between our classification and

GIEMS. Using GIEMS as a reference, the interannual variability of the 2000–2020 time series correlates strongly with our results (Pearson $r = 0.8540$, $p = 1.6659 \times 10^{-6}$; Spearman $\rho = 0.7729$, $p = 6.4580 \times 10^{-5}$). This high agreement provides supporting evidence that the post-2000 data are largely not biased by cloud contamination.

Given the requirement for concision in the Nature Communications main text and because cloud contamination is a methodological/data-processing issue, we have relocated the complete discussion of cloud effects to the Methods and expanded it in the Supplementary (Supplementary Discussion 3, Wetland area time series). The revised details can be found in the manuscript on [Page 18, lines 02 - 18]:

In this study, the wetland remote sensing classification and mapping work in the historical period of Africa were mainly divided into four parts: image preprocessing, wetland sampling, feature construction, and classification mapping (more details in the Supplementary Information).

Remote-sensing image preprocessing included standard cloud-and-shadow masking and noise removal. We applied robust annual compositing (e.g., multi-temporal median composites) to suppress short-term cloud contamination. To evaluate the effectiveness of these procedures, we compared our outputs against independent, cloud-insensitive reference products (for example, GIEMS). These comparisons indicate that, owing to the substantial improvement in Landsat data coverage and revisit frequency after 2000 and to our compositing plus multi-dataset validation strategy, the

wetland time series constructed for the period after 2000 show no evidence of a systematic bias attributable to cloud contamination. By contrast, data before 2000 are affected by larger image gaps and persistent cloud cover, resulting in reduced sample size and representativeness and therefore greater uncertainty in quantitative trend estimates. Relevant comparison figures and statistical summaries are provided in the Supplementary Information.

Major point 4

The authors have responded to most of my comments, but I am still unclear if the lack of fluviially-driven information implicit in FLDAS within the climate change simulations means that here is an important missing hydrological processes in some catchments.

Response: Is the reviewer referring to whether, in the future climate simulations, the absence of FLDAS (soil moisture reanalysis assimilation data) means that the model omits fluviially-driven hydrological processes?

We thank the reviewer for raising this important follow-up question. We agree that in the climate change (future period) simulations, the lack of FLDAS or other reanalysis-based soil moisture assimilation prevents explicit representation of river-driven hydrological processes. During the historical period, wetland dynamics benefited from the FLDAS soil moisture fields, which implicitly contained signals of river inflow and floodplain inundation through data assimilation. However, in the climate change (future) projections, the diagnostic model relies solely on GCM-derived meteorological variables (soil moisture) without such assimilation. Therefore, the simulated wetland dynamics mainly reflect local soil water balance and topographic control.

We thus acknowledge that some important hydrological processes—particularly river routing and upstream runoff contributions—are not represented in these future simulations. This omission may lead to an underestimation of variability in river-fed floodplain wetlands such as the Sudd and the Niger Inland Delta. We have revised the manuscript (Supplementary Information – Discussion) to clarify this limitation and emphasize that future work should couple the diagnostic framework with hydrological routing or surface water models to better capture fluviially-driven processes in such systems. The revised details can be found in the Supplementary Information on [Page 08 – 09]:

6. Effectiveness and uncertainty of inland wetland simulation

The classical TOPMODEL is a semi-distributed rainfall–runoff model based on the Topographic Wetness Index (TWI), originally proposed by Beven and Kirkby (1979). It was developed to simulate soil saturation and surface runoff processes under precipitation recharge. The model uses time series inputs of precipitation (along with other meteorological variables such as evapotranspiration and temperature) and estimates groundwater table depth (WTD) and runoff by calculating infiltration and saturation-excess overland flow, thereby enabling dynamic simulation of wetland extent in response to hydrological changes.

In the TOPMODEL-based diagnostic model employed in Xi et al. (2022) and in this study, precipitation time series is not directly used as the driving input. Instead, multiple reanalysis soil moisture (SM) datasets are used (in place of precipitation), together with soil temperature and freeze–thaw status, to calculate the average water table depth (WTD) for each grid cell. Subsequently, based on the analytical relationship between the Compound Topographic Index (CTI), which reflects topographic control, and the WTD, the inundation threshold CTI^* is derived. This allows for the diagnosis of the proportion of inundated/saturated pixels (on a monthly basis) within each grid cell, yielding the wetland fraction f_x , and thus enabling simulation of the spatiotemporal dynamics of wetlands.

In other words, this diagnostic algorithm "decouples" the explicit simulation of precipitation recharge in the original TOPMODEL by implicitly incorporating precipitation forcing into soil moisture reanalysis data, rather than directly treating precipitation as an independent driver. This approach is feasible for two main reasons:

(1) Land Data Assimilation Systems (LDAS) such as FLDAS indirectly account for the effects of runoff and river inflow on wetland water levels by assimilating remotely sensed soil moisture observations. As a result, the reanalyzed Water Table Depth (WTD) reflects variability driven by non-local precipitation.

(2) Although transient inundation driven by precipitation (e.g., flooded grasslands after heavy rainfall) falls under the broad definition of wetlands, its signal tends to be "averaged out" by short-term fluctuations in multi-decadal time series analysis. The model instead aims to capture long-term saturated zones that are more ecologically

significant in shaping ecosystem patterns. Thus, using soil moisture reanalysis instead of direct precipitation inputs not only simplifies computation but also emphasizes persistent wetland dynamics.

[Page 11-12]:

Additionally, during the historical period, the TOPMODEL-based diagnostic framework benefited from the FLDAS soil moisture fields, which implicitly incorporated signals of river inflow and floodplain inundation through data assimilation. However, this assimilation capability is unavailable in the climate change (future) simulations. The diagnostic model is driven solely by meteorological forcing (soil moisture) derived from global climate models (GCMs), and thus mainly reflects local soil water balance and topographic control. Consequently, fluvially-driven hydrological processes—particularly river routing and upstream runoff contributions—are not explicitly represented in these future simulations. This limitation may lead to an underestimation of variability in river-fed wetlands such as the Sudd and the Niger Inland Delta. Although the TOPMODEL-based diagnostic approach effectively captures long-term wetland dynamics governed by local soil moisture and terrain, it does not explicitly account for river-driven inundation. We recognize this as an important source of uncertainty and emphasize that future studies should couple the diagnostic framework with dynamic river routing or surface water models to better represent fluvial processes under changing climate conditions.

Furthermore, we acknowledge that while the TOPMODEL-based diagnostic model efficiently utilizes the compound topographic index (CTI) and reanalyzed soil moisture (SM) to estimate long-term saturation and inundation fractions at the grid scale, it does omit explicit simulations of river routing and floodplain processes, nor does it directly incorporate precipitation time series as input. Additionally, the model cannot resolve microtopography and small-scale wetland dynamics, nor does it account for anthropogenic regulation (e.g., dams, irrigation). Consequently, it exhibits notable limitations in simulating floodplains primarily driven by riverine inputs or extreme precipitation (e.g., the Sudd Wetland). Future improvements should involve coupling with river routing models, employing higher-resolution DEMs, and integrating direct

precipitation forcing.

Finally, we thank the reviewers again for their constructive comments. In response, we have incorporated the requested additions and revisions in the main text and in the Supplementary Information. We also agree that the reviewers' comments and the authors' response to the revised manuscript be made public upon online publication.

Reviewer #4 (Remarks to the Author)

General Comments: I'd like to thank the authors for responding fully to my original comments and incorporating elements of the response into the main text, methods and SI. I think they have successfully done that without disrupting the logical flow and need for brevity in a Nature Communications article. I recommend the manuscript for publication.

For information, the answer to the question raised by the authors in response to my "major point 4" (below) is yes.

"Is the reviewer referring to whether, in the future climate simulations, the absence of FLDAS (soil moisture reanalysis assimilation data) means that the model omits fluvially-driven hydrological processes?"

General response: The authors thank Reviewer 4 for the careful reading and positive recommendation. We appreciate the confirmation regarding Major Point 4. The manuscript, Methods and Supplementary Information have been updated to reflect this clarification, as noted in our point-by-point response. No further changes are required from the authors. Thank you for your constructive evaluation.